# The kinesin-5 tail domain directly modulates the mechanochemical cycle of the motor domain for anti-parallel microtubule sliding

Tatyana Bodrug[1‡], Elizabeth M Wilson-Kubalek[2†], Stanley Nithianantham[1†], Alex F Thompson[3†], April Alfieri[4†], Ignas Gaska[4], Jennifer Major[5,6], Garrett Debs[7], Sayaka Inagaki[6], Pedro Gutierrez[1], Larisa Gheber[8], Richard J McKenney[1], Charles Vaughn Sindelar[7], Ronald Milligan[2], Jason Stumpff[3], Steven S Rosenfeld[5,6], Scott T Forth[4], Jawdat Al-Bassam[1]*

[1]Department of Molecular and Cellular Biology, University of California, Davis, Davis, United States; [2]Department of Integrative Structural and Computational Biology, Scripps Research Institute, La Jolla, United States; [3]Department of Molecular Physiology and Biophysics, University of Vermont, Burlington, United States; [4]Department of Biological Sciences, Rensselaer Polytechnic Institute, Troy, United States; [5]Department of Cancer Biology, Lerner Research Institute, Cleveland Clinic, Lorain, United States; [6]Department of Pharmacology, Mayo Clinic, Jacksonville, United States; [7]Department of Molecular Biophysics and Biochemistry, Yale University, New Haven, United States; [8]Department of Chemistry and Ilse Katz Institute for Nanoscale Science and Technology, Ben-Gurion University of the Negev, Negev, Israel

*For correspondence: jmalbassam@ucdavis.edu

†These authors contributed equally to this work

Present address: ‡Department of Biochemistry and Biophysics, Lineberger Comprehensive Cancer Center, University of North Carolina, Chapel Hill, United States

Competing interests: The authors declare that no competing interests exist.

**Abstract** Kinesin-5 motors organize mitotic spindles by sliding apart microtubules. They are homotetramers with dimeric motor and tail domains at both ends of a bipolar minifilament. Here, we describe a regulatory mechanism involving direct binding between tail and motor domains and its fundamental role in microtubule sliding. Kinesin-5 tails decrease microtubule-stimulated ATP-hydrolysis by specifically engaging motor domains in the nucleotide-free or ADP states. Cryo-EM reveals that tail binding stabilizes an open motor domain ATP-active site. Full-length motors undergo slow motility and cluster together along microtubules, while tail-deleted motors exhibit rapid motility without clustering. The tail is critical for motors to zipper together two microtubules by generating substantial sliding forces. The tail is essential for mitotic spindle localization, which becomes severely reduced in tail-deleted motors. Our studies suggest a revised microtubule-sliding model, in which kinesin-5 tails stabilize motor domains in the microtubule-bound state by slowing ATP-binding, resulting in high-force production at both homotetramer ends.

## Introduction

Microtubules (MTs) form tracks for the active transport of vesicles and macromolecules inside eukaryotic cells, generate pulling forces during assembly of mitotic spindles, and promote the alignment and segregation of chromosomes (*Goshima and Scholey, 2010*; *Vale, 2003*). Fourteen kinesin motor subfamilies utilize MTs as tracks for many diverse functions (*Vale, 2003*). Among them, kinesin-5 motors represent a unique and highly conserved subfamily that is essential for mitotic spindle assembly during metaphase and spindle elongation during anaphase (*Kashina et al., 1996*). In

contrast to the majority of kinesin classes, kinesin-5 motors adopt a conserved bipolar homotetrameric organization, composed of two dimeric subunits folded in an antiparallel arrangement mediated by the assembly of a 60 nm long central minifilament (*Acar et al., 2013*; *Kashina et al., 1996*; *Scholey et al., 2014*; *Singh et al., 2018*). Through this conserved bipolar organization, kinesin-5 motors promote MT crosslinking and mediate their sliding apart during mitotic spindle assembly and elongation. This activity can be recapitulated in vitro with purified kinesin-5 motors from a variety of species (*Kapitein et al., 2008*; *Kapitein et al., 2005*; *van den Wildenberg et al., 2008*).

Metazoan kinesin-5 motors such as *D. melanogaster* KLP61F or human Eg5 exhibit slow plus-end directed motility especially during antiparallel MT sliding (*Kapitein et al., 2008*; *Kapitein et al., 2005*; *Shimamoto et al., 2015*; *van den Wildenberg et al., 2008*). In contrast, yeast kinesin-5 motors, such as Cin8, Kip1 and Cut7 uniquely undergo minus-end directed motility as single motors and reverse direction toward MT plus-ends upon clustering into multi-motor assemblies along single MTs, or during antiparallel MT sliding (*Edamatsu, 2014*; *Fridman et al., 2013*; *Gerson-Gurwitz et al., 2011*; *Roostalu et al., 2011*; *Shapira et al., 2017*). The conserved plus-end directed MT sliding activity is essential for mitotic spindle assembly by generating forces exerted on MTs emanating from opposite spindle poles during metaphase and stabilizing the characteristic bipolar spindle organization (*Brust-Mascher et al., 2009*; *Forth and Kapoor, 2017*; *Subramanian and Kapoor, 2012*; *Wang et al., 2014*). The MT sliding activity is critical for the elongation of mitotic spindles at the midzone region during anaphase (*Goshima and Scholey, 2010*). Defects in mammalian kinesin-5 motors or their inactivation via inhibitory compounds such as monastrol disrupt the balance of mechanical forces within the spindle and lead to monopolar spindles (*Goshima and Scholey, 2010*; *Goshima et al., 2005*). These inhibitory compounds aided in elucidating the fundamental functions of kinesin-5 in mitosis and were suggested to be of therapeutic value in treating rapidly dividing cancer cells (*Kwok et al., 2006*; *Mayer et al., 1999*; *Owens, 2013*).

Each kinesin-5 motor consists of a conserved organization including: an N-terminal motor domain, α-helical neck and bipolar assembly regions, and a C-terminal tail domain. The motor domain is connected via a neck-linker to a dimerizing α-helical coiled-coil neck (*Turner et al., 2001*; *Valentine et al., 2006a*; *Valentine et al., 2006b*). The parallel α-helical coiled-coil neck forms part of the 60 nm central antiparallel homotetrameric α-helical minifilament (*Acar et al., 2013*; *Kashina et al., 1996*). At the center of this minifilament is a 27 nm antiparallel four α-helical bundle termed the bipolar assembly (BASS) region (*Scholey et al., 2014*). The bipolar tetrameric organization of the kinesin-5 BASS region orients the two-parallel coiled-coils at the neck and their associated motor domains at an off-set. This results in a 100°-lateral rotation between the two opposite ends that potentially mediates the preference for kinesin-5 to bind and slide two antiparallel MTs (*Scholey et al., 2014*). A section of unknown structure extends from the C-terminus of the BASS to the tail domain that is located near the motor domains of the antiparallel subunits (*Acar et al., 2013*). Thus, each end of the kinesin-5 tetramer consists of twin tail and motor domains that emerge from intertwined antiparallel dimeric subunits (*Acar et al., 2013*; *Scholey et al., 2014*).

The kinesin-5 tail domain contains a conserved BimC box, which is a consensus motif that is phosphorylated by mitotic cyclin-dependent kinases (*Blangy et al., 1995*; *Sharp et al., 1999*). Mitotic phosphorylation at the BimC box induces kinesin-5 motors to concentrate along the mitotic midzone where they promote the elongation of the mitotic spindle during late anaphase by sliding apart antiparallel MTs (*Sharp et al., 1999*), though the role for this phosphorylation in the regulation of kinesin-5 activity remains unknown. Studies of the *S. cerevisiae* yeast ortholog Cin8 show the tail domain is essential for kinesin-5 function; deletion of the tail leads to a lethal mitotic arrest phenotype in the absence of the analogous motor Kip1 (*Hildebrandt et al., 2006*). The tail domain of the *Xenopus laevis* Eg5 has been suggested to form a secondary MT binding site during MT sliding motility (*Weinger et al., 2011*). However, the exact function of the kinesin-5 tail domain remains unknown. Despite extensive structural, kinetic, and functional analyses of kinesin-5 motors, the origin of the highly conserved MT sliding activity in these motors and its relation the conserved tetrameric organization remains poorly understood.

Here, we describe a mechanism for the regulatory motor-to-tail interaction within the homotetrameric kinesin-5 motor and its fundamental role during MT sliding motility. Using biochemical methods, we show that the kinesin-5 tail domain decreases MT-stimulated ATP hydrolysis by binding and stabilizing the motor domain in its ADP or nucleotide-free states. Cryo-EM structures reveal that the KLP61F tail stabilizes the open conformation of the motor by binding its N-terminal subdomain via

the α0-helix element located at its tip. We show that human Eg5 motors undergo very slow motility and form clusters along individual MTs, whereas deletion of the tail domain causes Eg5 to undergo rapid motility without clustering along individual MTs. Single-motor fluorescence tracking in MT sliding assays show that Eg5 motors undergo slowed unidirectional motility within active sliding zones, while tail-deleted Eg5 motors undergo rapid motility with frequent switches in direction along the antiparallel MTs within sliding zones. Optical trapping and MT sliding assays reveal that the tail is essential for zippering two MTs into sliding zones through its capacity to generate high pushing forces. In cells, tail-deletion leads to a loss of human Eg5 mitotic spindle localization in mammalian cells while retaining MT binding capability. These studies suggest a revised model for kinesin-5-mediated MT sliding in which the tail domain slows MT-stimulated ATP hydrolysis at each end of the homotetramer and enhance force production essential for MT sliding.

## Results

### The kinesin-5 tail domain decreases motor domain MT-stimulated ATP hydrolysis

To better understand the role of the tail domain on kinesin-5 function, we first evaluated the effect of the tail domain on MT-stimulated ATP hydrolysis by the kinesin-5 motor domain. To circumvent the added complexity of MT crosslinking and sliding of the full length protein (*Figure 1A*), we generated constructs consisting of the motor-neck-linker domains (termed 'motor'; residues 1–356), the isolated tail (termed 'tail'; residues 913–1056), and a fusion construct in which the motor-neck-linker is linked at its C-terminus to the tail via an 8-residue linker (termed 'motor-tail fusion') (*Figure 1B–D*). We generated and studied both human-Eg5 and Dm-KLP61F constructs in parallel to characterize the conservation of features across these two well-studied orthologs (*Table 1*; *Figure 1B–D*; *Figure 1—figure supplement 1K–M*). For these assays, increasing levels of MTs were added to the motor, to a mixture of motor and tail, or to the motor-tail fusion and resulting MT stimulated ATP hydrolysis rates were measured (Materials and methods; *Figure 1B–D*). The KLP61F motors robustly hydrolyzed ATP in response to increasing amounts of MTs ($k_{cat}$7.1 s$^{-1}$ and $K_m$680 nM; *Figure 1B*; *Table 1*). The addition of equimolar KLP61F tail to a mixture of the motor and MTs led to a two-fold decrease in $k_{cat}$ (3.5 s$^{-1}$ vs 7.1 s$^{-1}$) but produced little change in $K_m$ (756 nM vs 680 nM) suggesting that the tail does not competitively interfere with motor-MT binding but rather modulates MT-stimulated ATP hydrolysis (*Figure 1C*). The motor-tail fusion exhibited a 2.2-fold decrease in $k_{cat}$ compared to the motor alone (3.4 s$^{-1}$ vs 7.1 s$^{-1}$) and a 15-fold decrease in $K_m$ compared to that for the motor alone (39 nM vs 680 nM) (*Figure 1D*). We observed similar behavior for human Eg5 motor and tail constructs. Although, the Eg5 motor displays an eight-fold increase in $K_m$ compared to KLP61F motor at 50 mM K-acetate condition (3849 nM vs 680) suggesting the Eg5 motor is slightly sensitive to ionic strength. Therefore, we studied Eg5 constructs at both 20 mM KCl and 50 mM K-acetate ionic strength conditions. At 20 mM KCl condition, the Eg5 tail decreases the MT stimulated Eg5 motor ATP hydrolysis rate by 32% compared to Eg5 motor alone ($K_{cat}$7.6 s$^{-1}$ vs 5.3 s$^{-}$'; *Table 1*; *Figure 1—figure supplement 1K*). The decrease in homogeneous solubility of the Eg5 tail construct compared to the KLP61F tail is likely responsible for this mild decrease in MT-stimulated ATPase. An Eg5 motor-tail fusion construct exhibited a two-fold decrease in MT-stimulated ATPase $K_{cat}$ (3.47 s$^{-1}$ vs 7.55 s$^{-1}$) nearly identical to a parallel construct for Klp61f (*Figure 1—figure supplement 1L*). The tail-induced decrease in MT-stimulated ATPase activity, however, was somewhat reduced at a 50 mM K-acetate condition (5.71 s$^{-1}$ vs 6.81 s$^{-1}$; *Table 1*; *Figure 1—figure supplement 1M*), suggesting that in this condition the tail-to-motor regulation is weakened (see next section). Our studies reveal that the kinesin-5 tail domain decreases the MT-stimulated ATP hydrolysis rate of the motor domain in either an isolated or a fused configuration and that this feature is conserved across human Eg5 and Dm-KLP61F.

### The tail domain binds the motor domain in the ADP or nucleotide-free states

To understand the biochemical basis for the kinesin-5 tail-mediated regulation of motor domain MT-stimulated ATP hydrolysis, we studied the binding of kinesin-5 motor domain with and without the MT lattice. First, we measured the capacity of the motor domain to co-purify and co-elute with a

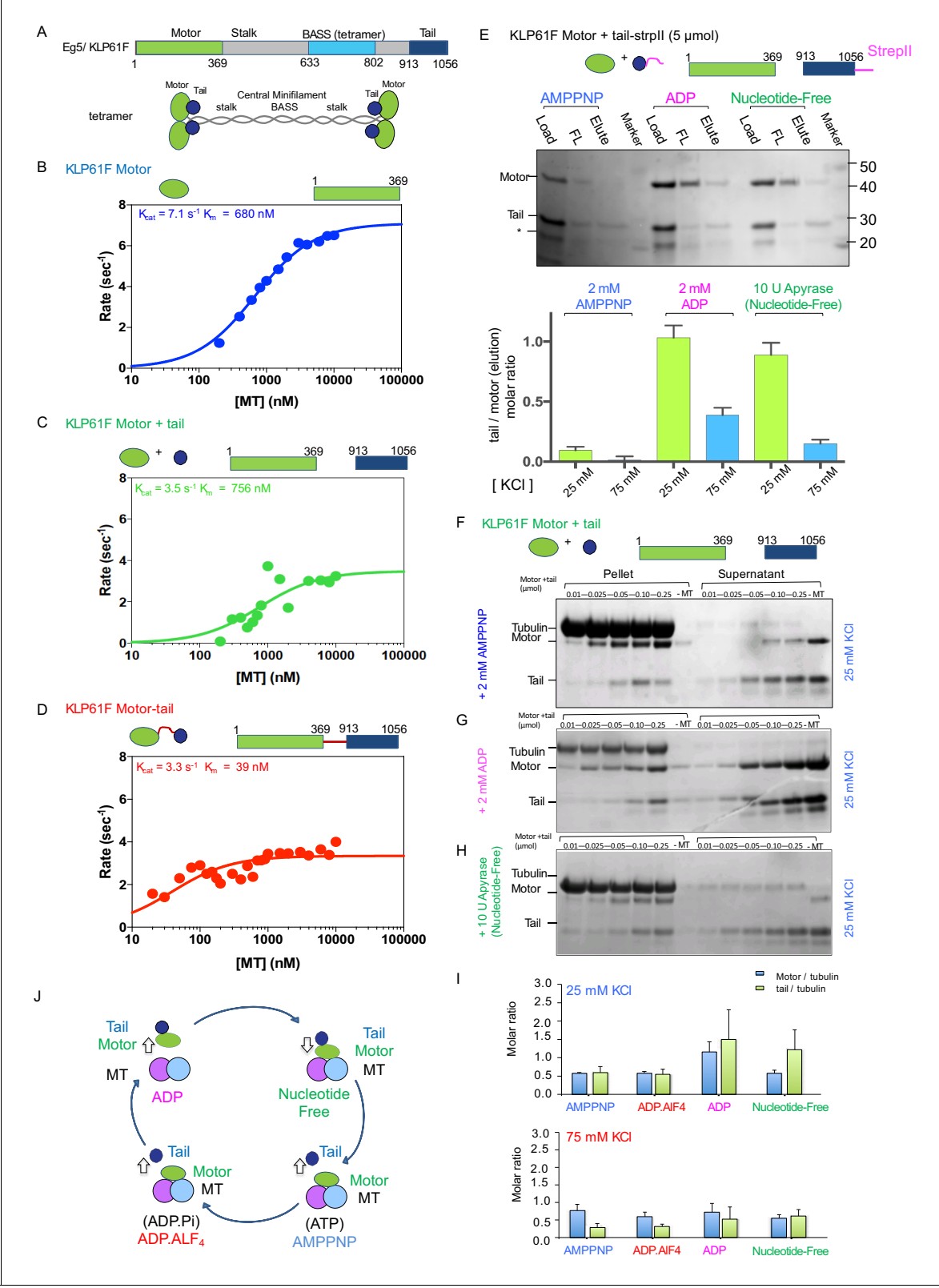

**Figure 1.** The kinesin-5 tail domain inhibits the motor domain MT-stimulated ATPase activity through stabilization of the MT bound nucleotide-free state. (**A**) Top, domain organization for the kinesin-5 motors, Dm KLP61F and Hs FL-Eg5 consisting of conserved N-terminal motor domain, central BASS domain and C-terminal tail domain. Bottom, homotetrameric organization of kinesin-5. (**B**) Steady-state MT-stimulated ATP hydrolysis for KLP61F motor. (N = 2) (**C**) Steady state MT stimulated ATP hydrolysis for equimolar KLP61F motor + tail constructs. (N = 2) (**D**) Steady state MT stimulated ATP

*Figure 1 continued on next page*

*Figure 1 continued*

hydrolysis for KLP61F motor-tail fusion. (N = 2) (**E**) Affinity co-purification of the KLP61F motor using the KLP61F tail-StrepII trapping one of three nucleotide states conditions. Top panel, SDS-PAGE for each condition: 5 µmol motor + tail mixture is loaded onto Streptactin XT resin (Load), flow-through fraction (FL) and biotin elution fraction (elute) are shown. Left lanes, incubation of motor + tail with non-hydrolysable ATP-analog, 2 mM AMPPNP. Center lanes, incubation of motor + tail with 2 mM ADP. Right lanes, incubation of motor + tail in the presence of 2 U Apyrase resulting in a nucleotide free state. Bottom panel, quantitative densitometry reveals the molar ratios of motor to tail in the elution fractions at 25 mM KCl (yellow column) and 75 mM KCl (blue columns) buffer conditions (N = 3, n = 6 for each column). (**F**) MT co-sedimentation assays of the KLP61F motor (motor) and tail domain (tail) with MTs (MT) in the presence of non-hydrolysable analog, 2 mM AMPPNP at 25 mM KCl. Co-sedimentation was carried out with MTs for 0.01, 0.025, 0.05, 0.1,0.25 µmol motor and tail mixture at 25 mM KCl conditions, and in the absence of MTs (-MT) control. Additional data are shown in *Figure 1—figure supplement 1B*. (**G**) MT co-sedimentation assays of the KLP61F motor (motor) and tail domain (tail) with MTs (MT) in the presence of 2 mM ADP at 25 mM KCl. Co-sedimentation was carried out with MTs for 0.01, 0.025, 0.05, 0.1,0.25 µmol motor and tail mixture at 25 mM KCl condition and in the absence of MTs (-MT) control. Additional data are shown in *Figure 1—figure supplement 1D*. (**H**) MT co-sedimentation assays of the KLP61F motor (motor) and tail domain (tail) with MTs (MT) in the nucleotide-free state, achieved by adding 10 U of Apyrase. Co-sedimentation was carried out with MTs for 0.01, 0.025, 0.05, 0.1,0.25 µmol motor and tail mixture at 25 mM KCl conditions and in the absence of MTs (-MT) control. Detailed and additional data are shown in *Figure 1—figure supplement 1E*. (**I**) Molar ratios of motor to tubulin monomer (blue) and tail to tubulin dimer (green) measured using quantitative densitometry (Materials and methods). The motor-to-tubulin ratios observed at 75 mM KCl are roughly 0.5 and the amount of tail bound to the motor increases in the ADP and Nucleotide-free states increases in the 25 mM KCl compared to the 75 mM KCl conditions. The amount of tail bound to motor remain low in the AMPPNP and ADP.AlF4 states compared to the ADP and nucleotide-free states. (**J**) Summary of data presented resulting in a model for motor (green) and tail (blue) domain affinities and MT (blue and pink) binding capacities in four nucleotide states during the ATP hydrolysis cycle.

The online version of this article includes the following figure supplement(s) for figure 1:

**Figure supplement 1.** The kinesin-5 tail domain inhibits motor domain MT-stimulated ATPase activity through stabilization of ADP-bound and nucleotide-free state.

C-terminally strep-II tagged KLP61F tail domain construct onto streptactin XT resin (See Materials and methods; *Figure 1E*; *Figure 1—figure supplement 1A*). We compared the binding activities at two ionic strength conditions (25 and 75 mM KCl) for which motility assays indicated distinct modes of tail-dependent motility regulation (see below; Figures 3–4). We also tested how the binding is influenced by distinct nucleotides that trap the motor domain in different steps of the ATP hydrolysis cycle (*Figure 1E*). At 25 mM KCl, the motor co-eluted with the tail in the 2 mM ADP and in the nucleotide free state (in presence of Apyrase enzyme) but very poorly co-eluted with the tail in the presence of 2 mM AMPPNP (*Figure 1E*). Furthermore, the proportion of Klp61F motor co-eluting with the tail decreased dramatically at 75 mM KCl conditions (*Figure 1E*). Higher concentration of motor and tail did not change the molar ratio of the eluting proteins (*Figure 1—figure supplement 1A*).

Next, we reconstituted binding of the Klp61F and Eg5 motor and tail constructs onto Paclitaxel-stabilized MTs in different nucleotide state conditions and analyzed the binding using MT co-sedimentation assays (*Figure 1F–I*; *Figure 1—figure supplement 1B–H*). We again used conditions that mimic each step of the kinesin ATP hydrolysis cycle (*Figure 1F–I*; *Figure 1—figure supplement 1B–H*). We also compared the binding activities at two ionic strength conditions (25 and 75 mM KCl) as described above. The KLP61F tail and motor individually bound MTs in the presence of 2 mM ADP, as has been shown previously (*Figure 1—figure supplement 1G–H*; *Weinger et al., 2011*). The

**Table 1.** Steady kinetic parameters for MT-activated ATP hydrolysis.

| Construct | Source | Ionic Strength | $k_{cat}$ (sec$^{-1}$) | $K_{0.5,MT}$ (nM) |
|---|---|---|---|---|
| Motor | *Dm KLP61F* | 50 mM K Acetate | 7.1 ± 0.1 | 680 ± 48 |
| Motor + Tail | *Dm KLP61F* | 50 mM K Acetate | 3.5 ± 0.5 | 757 ± 327 |
| Motor-Tail fusion | *Dm KLP61F* | 50 mM K Acetate | 3.3 ± 0.1 | 39 ± 11 |
| Motor | Hs Eg5 | 20 mM KCl | 7.3 ± 0.2 | 334 ± 14 |
| Motor-Tail fusion | Hs Eg5 | 20 mM KCl | 3.4 ± 0.2 | 158 ± 41 |
| Motor + Tail | Hs Eg5 | 20 mM KCl | 5.4 ± 0.3 | 209 ± 56 |
| Motor | Hs Eg5 | 50 mM K Acetate | 6.71 ± 0.7 | 3849 ± 800 |
| Motor-Tail fusion | Hs Eg5 | 50 mM K Acetate | 5.87 ± 0.23 | 391 ± 59 |

molar ratio of the bound tail and motor to MTs (polymerized tubulin) are roughly ~0.5 and~1 per αβ-tubulin at 25 mM KCl and decreased at 75 mM KCl (*Figure 1—figure supplement 1F,H*). MT co-sedimentation of motor and tail together in the presence of 2 mM AMPPNP or ADP.AlF4 revealed that the tail is displaced from MTs, while the motor bound with high affinity to saturation (*Figure 1F*; *Figure 1—figure supplement 1B–C*). Even at the lowest molar ratio of motor to MTs, where there is an abundance of unoccupied MT lattice sites, the tail does not co-sediment with MTs and does not compete with the motor for MT binding sites.

In the ADP and nucleotide free states, the KLP61F motor domain recruits the tail domain more effectively to the MT lattice despite the different motor affinities for MTs. In the presence of 2 mM ADP, the motor binds MTs with low affinity in a concentration-dependent manner, and its binding molar ratio is higher at 25 mM KCl compared to 75 mM KCl (*Figure 1G*; *Figure 1—figure supplement 1D*). At 25 mM KCl, a higher amount of motor binds MTs, resulting in a higher amounts of tail being recruited into the pellet, approaching a molar ratio of ~1 motor per αβ-tubulin MT lattice site (*Figure 1G*; *Figure 1—figure supplement 1D*). At 75 mM KCl, both motor and tail amounts bound to MTs decreased and their molar ratio decreased (*Figure 1I*). MT co-sedimentation in the presence of apyrase (1 U/ml) to promote thenucleotide-free state of the motor revealed a high affinity of the motor for MTs at both 25 and 75 mM KCl (*Figure 1H*). Under the nucleotide-free condition, higher levels of tail are recruited in a manner that correlates with the amount of motor bound to MT in the pellet fraction (*Figure 1I*). Quantitative densitometry analyses indicates the molar ratios of tail recruited to the MT-bound fraction are highest in the ADP and nucleotide-free state in contrast to the AMPPNP and ADP.AlF4 states (*Figure 1I*). We studied analogous human Eg5 motor and tail constructs in the AMPPNP, ADP, and nucleotide free states which revealed essentially similar patterns of behavior to those described above for KLP61F (*Figure 1—figure supplement 1I–J*). The human Eg5 motor recruits the Eg5 tail to the MT-bound fraction in the presence of 2 mM ADP. Despite the high affinity of the Eg5 motor for MTs in the presence of 2 mM AMPPNP, the tail remains unbound and in the soluble fraction (*Figure 1—figure supplement 1H–J*). This suggest that the Eg5 tail binds to Eg5 motor in the presence of 2 mM ADP, but its binding to the Eg5 motor in presence of 2 mM AMPPNP is similar but weaker than the KLP61F tail (*Figure 1—figure supplement 1I–J*). Together, our studies revealed that the ATP and ADP.Pi states of the kinesin-5 motor domains exhibit a high binding affinity for MTs, but a low affinity for the tail domain, while also displacing the isolated tail from MTs. In contrast, in the ADP and nucleotide-free states the kinesin-5 motor exhibits increased affinity for the tail, leading the tail to be recruited to the MT bound fraction despite the difference in the motor-MT affinity in these two states (*Figure 1J*).

## Cryo-EM structure reveals the tail domain stabilizes an open active site conformation of the motor domain

We used cryo-electron microscopy (cryo-EM) to investigate the kinesin-5 tail-motor interface and its role in motor mediated ATP hydrolysis. We collected cryo-EM images of KLP61F motor decorated onto MTs in the AMPPNP state, and the KLP61F motor and tail decorated onto MTs in the nucleotide free state. Class averages for the MT-decorated segments in the two states clearly revealed repeating motordensities decorating the MT lattice sites (*Figure 2E*). New and previously described image analysis strategies (see Materials and methods) were used to calculate and refine structures of MTs decorated with the KLP61F motor in the AMPPNP state and the KLP61F motor and tail complex in the nucleotide-free state to ~4.4 Å. We adopted a new strategy to apply local classification to the motor and tail densities in the nucleotide free state, which involved classifying small patch of the MT lattice (Materials and methods). Only 70% of the data were utilized for this refinement step (Materials and methods). However, this new strategy generally improved the resolution of the motor domain density but did not enhance the occupancy or resolution for the tail (*Figure 2A–F*; *Figure 2—figure supplement 1A–C*; *Table 2*; see Materials and methods). Additional density, which we attribute to the tail domain, is observed on top of the motor domain in the nucleotide-free motor and tail-decorated MTs. Despite the size of the tail domain (80–100 residues), this density is small and accounts for only about one third of this mass (*Figure 2—figure supplement 1B–C,H*). This suggests that two thirds of the tail is either unstructured, or forms a flexibly attached and separate domain; a major part of the tail region is rendered invisible after averaging. Furthermore, the density attributed to the tail region cannot be interpreted in terms of secondary structure as there is a substantial drop in the resolution to ~8 Å at the motor-tail interface (*Figure 2—figure supplement 1B–*

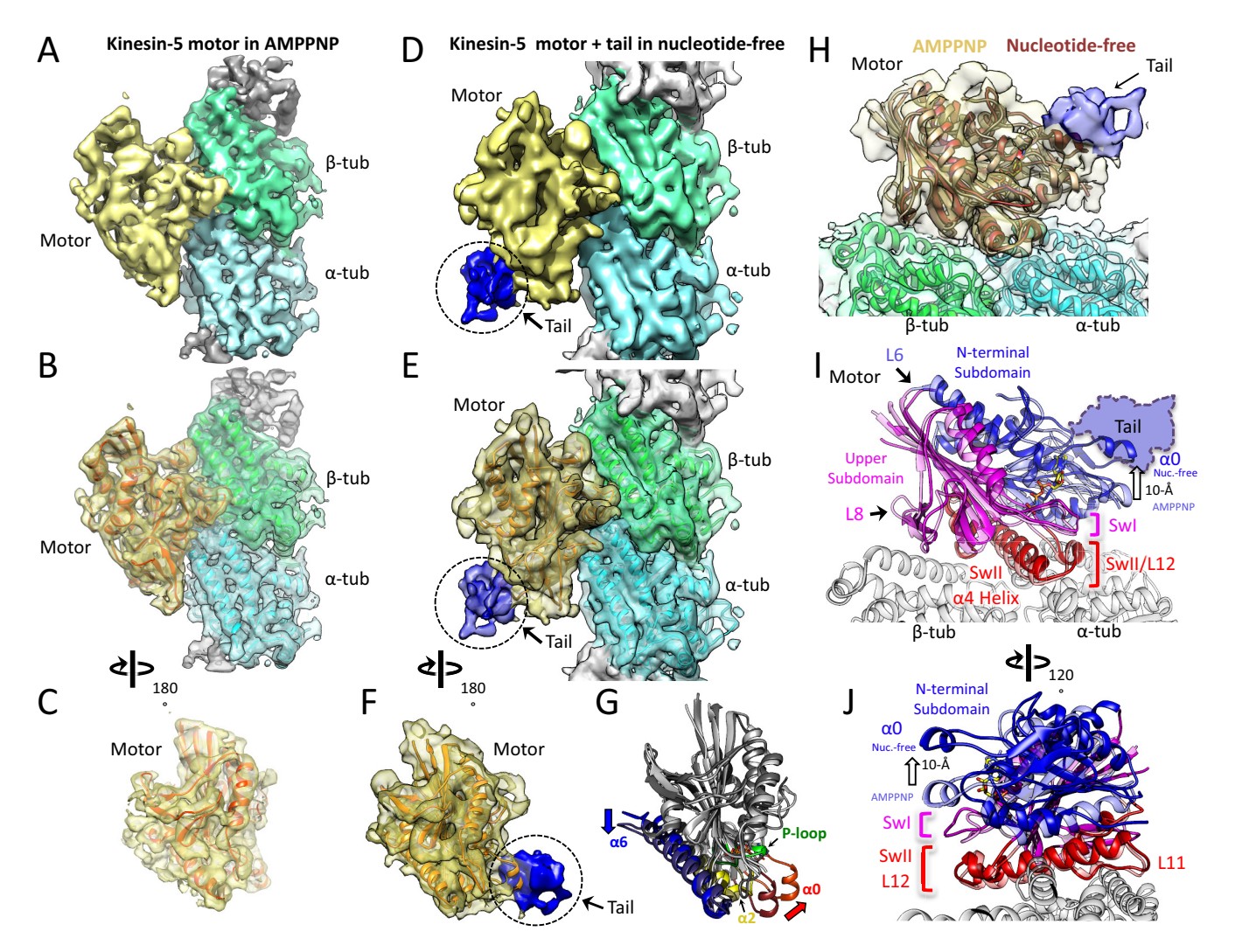

**Figure 2.** Cryo-EM reveals the kinesin-5 tail engages the motor domain in the nucleotide-free state and stabilizes an open ATP active site. (A-C) (A) Side view of 4.0 Å cryo-EM structure of Klp61F motor domain decorated MT unit in the AMPPNP state. A single kinesin motor-bound αβ-tubulin unit is shown. The segmented motor domain (yellow) and α-tubulin (cyan) and β-tubulin (green) densities are sown. (B) *De novo* built KLP61F motor domain model in the AMPPNP state (red) displayed within the motor domain density. The αβ-tubulin dimer model fitted into the αβ-tubulin density (green and cyan). (C) Top end view of the kinesin-5 motor density map with the MT density computationally removed. (D-F) (D) Side view of 4.0 Å cryo-EM structure of Klp61F motor and tail domains decorated onto MTs in the nucleotide-free state obtained using well-established and refined with new strategies (*Figure 2—figure supplement 1A–C*; see Materials and methods). The kinesin-motor + tail bound to a single αβ-tubulin unit is shown. Segmentation of the motor domain (yellow), tail domain is shown (blue), α-tubulin (cyan) and β-tubulin (green). (E) *De novo* built Klp61F motor domain model in the nucleotide-free state (orange) displayed within the motor domain density and the tail density (dark blue). The αβ-tubulin dimer fitted into the αβ-tubulin density (green and cyan). (F) Top-end view of the docked kinesin-5 motor and tail density with the MT density computationally removed. (G) Conformational transition of Klp61F motor domain from nucleotide-free (light gray) to AMPPNP (dark gray). The elements that undergo the most change in colors: α6 (blue), α2 (yellow), p-loop (green) and α0 helix (red). (I-J) Two views of the Klp61F AMPPNP and nucleotide-free motor domain models describing the movements of motor with N-terminal subdomain (deep blue, nucleotide-free (nuc.-free); light blue, AMPPNP) and Upper subdomain (deep pink, nucleotide-free; light pink, AMPPNP) around the α4-helix, L11 switch II (swII) MT bound subdomain (deep red, nucleotide-free; light red, AMPPNP). The switch I (swI) changes conformation in response to ATP binding. The N-terminal subdomain rotation leads to a 20˚ rotation of the α0-helix closer to the MT surface in the AMPPNP state. The tail binds the α0-helix in the nucleotide-free state. H) Side-view of the Klp61F motor domain maps in AMPPNP (red) compared to the nucleotide-free state model (orange) docked into the nucleotide free-state motor density with the tail density shown in blue.

The online version of this article includes the following figure supplement(s) for figure 2:

**Figure supplement 1.** Cryo-EM reveals the kinesin-5 tail domain engages the motor domain directly through the α0-helix and stabilizes an open ATP active site.

**Table 2.** Cryo-EM KLp61F motor and tail MT structures: collection and reconstruction.

| | Dm Klp61F motor-AMPPNP (15 protofilaments) | Dm Klp61F- motor AMPPNP (14 protofilaments) | Dm KLP61F motor-tail- nucleotide free (15 protofilaments) | Dm Klp61F5 motor-tail-nucleotide free (14-protofilaments) |
|---|---|---|---|---|
| **Data collection** | | | | |
| Microscope | Titan Krios (FEI) | Titan Krios (FEI) | Titan Krios (FEI) | Titan Krios (FEI) |
| Voltage (kV) | 300 | 300 | 300 | 300 |
| Ls | 22,500X | 22,500X | 22,500X | 22,500X |
| Cumulative exposure dose (e$^-$ Å$^{-2}$) | 38 | 38 | 40 | 40 |
| Exposure rate (e$^-$/pixel/sec) | 7.9 | 7.9 | 8.3 | 8.3 |
| Detector | K2 Summit | K2 Summit | K2 Summit | K2 Summit |
| Pixel size (Å)* | 1.31 | 1.31 | 1.31 | 1.31 |
| Defocus range (μm) | 0.3–3.78 | 0.7–3.78 | 0.19–5.12 | 0.19–5.12 |
| Average defocus (m) | 1.75 | 1.75 | 1.86 | 1.86 |
| Micrographs Used | 1260 | 1260 | 955 | 955 |
| Total extracted helical segment (no.) | 73,451 | 73,451 | 44,081 | 44,081 |
| Refined helical segment (no.) | 21,004 | 39,001 | 9490 | 27,433 |
| **Reconstruction** | | | | |
| Final helical segments (no.) | 21,004 | 39,220 | 9490 | 14,570 |
| Symmetry imposed | HP | HP | HP | HP |
| Resolution (global) FSC 0.143 | 4.2 | 4.4 | 4.2 | 4.3 |

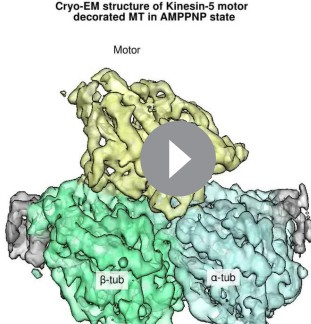

**Video 1.** Structural transition of the kinesin-5 motor domain from AMPPNP to nucleotide state and its effect on binding of the tail domain. View of the kinesin-5 motor domain map with AMPPNP showing the motor domain model, transition to the motor nucleotide state map showing the site of binding for the tail domain density, and model for the motor. Views of the two states using three motor subdomains and their conformational changes in the N-terminal subdomain (blue) and its effect on the ATP binding site and the rotation around the Upper subdomain (pink) and the MT bound subdomain (red).
https://elifesciences.org/articles/51131#video1

*C*; *Video 1*). In keeping with the biochemical data, we do not observe any ordered additional density on the MT lattice, consistent with the idea that the tail does not bind the MT lattice in the presence of the motor domain. Overall, the data strongly suggest that the tail interacts loosely with the underlying motor domain, binding directly to its α0 helix hairpin element, located on its MT minus end facing side (*Figure 2D–F*). Comparison of the KLP61F motor AMPPNP structure with the motor-tail nucleotide-free structure revealed conformational changes within the motor domain leading a major rotation in the α0 helix hairpin (*Figure 2D–F*; *Video 1*), and suggests how elements of the nucleotide-free state form an effective binding site for the tail (*Figure 2C*). One end of the α0 helix is connected to the phosphate binding loop (p-loop) (*Figure 2G*) and allows possible feedback between the nucleotide pocket and the tail binding site.

Model building for both the kinesin-5 AMPPNP and nucleotide-free motor domain structures revealed conformational changes in α0-helix hairpin that may regulate the affinity of the tail within the motor domain (*Table 3*; *Figure 2C,F*; *Video 1*). Compared to kinesin-1,

**Table 3.** Cryo-EM refinement and Structure model statistics.

| | Dm-Klp61F motor AMPPNP-MT | Dm-Klp61F motor -Nucleotide free-MT |
|---|---|---|
| **Data collection** | | |
| Microscope/detector | Titan Krios/Gatan K2 | Titan Krios/Gatan K2 |
| Magnification | 22,500x | 22,500x |
| Voltage (keV) | 300 | 300 |
| Dose rate (electrons/pixel/second) | 7.96 | 8.3 |
| Pixel size (A°/pixel) | 1.31 | 1.31 |
| Map resolution (A°) | 4.4 | 4.4 |
| FSC threshold | 0.143 | 0.143 |
| **Refinement** | | |
| Model resolution cutoff (A°) | 4.4 | 4.4 |
| FSC threshold | 0.143 | 0.143 |
| Protein residues | 1173 | 1174 |
| Ligands | 3 (GTP/GDP/AMPPNP) | 2 (GTP/GDP) |
| Map CC | 0.8 | 0.78 |
| B factor (A°) | 216 | 208 |
| **R.M.S deviations** | | |
| Bond lengths (A°) | 0.003 | 0.006 |
| Bond angles (°) | 0.53 | 1.14 |
| **Validation** | | |
| All-atom clash score | 14.05 | 11.71 |
| MolProbity score | 2.52 | 1.9 |
| **Ramachandran plot** | | |
| Favored (%) | 96.19 | 94.65 |
| Allowed (%) | 3.81 | 5.18 |
| Outliers (%) | 0.00 | 0.17 |

the kinesin-5 motor structure revealed the conserved kinesin fold with two longer and reorganized class-specific L6 and L8 loops which form part of the motor MT-binding interface (*Figure 2—figure supplement 1E–F-G*), similar to the map of the recently-determined *S. pombe* and *U. maydis* kinesin-5 motor domain structures (*von Loeffelholz and Moores, 2019*; *von Loeffelholz et al., 2019*). Structurally, the kinesin-5 motor domain can be divided to three subdomains, comparable to those seen in kinesin-1: the N-terminal subdomain, the upper subdomain, and the MT-binding subdomain (*Figure 2FH-I*; *Cao et al., 2014*; *Shang et al., 2014*). The MT-bound subdomain consists of the α4-helix L11 and L12, as described for kinesin-1 (*Figure 2I–J*; *Figure 2—figure supplement 1E–F-G*; *Video 1*). As in kinesin-1, the N-terminal subdomain rotates around the MT-bound subdomain in the kinesin-5 nucleotide-free state, leading to a reorganization of the MT minus-end facing end of the motor (*Figure 2H–I*; *Figure 2—figure supplement 1F–G*). In the nucleotide-free state, the N-terminal subdomain (blue) rotates upward with respect to the upper subdomain (pink) and the MT bound α4 helix (switch-II), and L11 (red) to open the switch I loop in the active site (*Figure 2H–J*). Compared to the AMPPNP state, the N-terminal subdomain in the nucleotide-free state rotates upward by 20° causing the α0-helix hairpin, which lies at its extreme tip, to move upward by 10 Å from the MT surface (*Figure 2H–J*; *Video 1*). This N-terminal subdomain rotation repositions the switch I, switch II, and P-loops, resulting in an open active site (*Figure 2H–J*), whereas in the AMPPNP state, the α0-helix is positioned downward and 10 Å closer to the MT lattice, (*Figure 2H–J*) due to ATP binding and active site closure. Our structures suggest that binding of the tail domain to the N-terminal subdomain likely stabilizes the upward α0-helix conformation which prevents the P-loop and switch I//II from engaging ATP (*Figure 2J*; *Video 1*). Our models suggest that the tail interface

stabilizes this upward N-terminal subdomain motor conformation leading to a decrease in the ability of the active site to bind incoming ATP. This suggests that the motor-α0-helix may cycle between 'on' and 'off' states while binding the tail to result in a slower ATP hydrolysis cycle.

## The kinesin-5 tail domain decreases homotetramer velocity along single MTs

We next set out to examine the role of the tail to motor interface in kinesin-5 motility, we reconstituted the motility of homotetrameric human Eg5 motors along individual MTs. We studied two human Eg5 constructs that either contain or exclude the tail domain (*Figure 3A*). We purified full-length Eg5 (termed FL-Eg5, residues 1–1056), full-length Eg5 with a C-terminal GFP (termed FL-Eg5-GFP), and mutant Eg5 with the tail domain deleted with a C-terminal GFP (termed Eg5-Δtail-GFP, residues 1–912) (*Figure 3A–B*). These motors were expressed in insect cells and purified using StrepII-tag affinity, followed by size exclusion chromatography (*Figure 3—figure supplement 1A–B*; see Materials and methods). Deleting the tail domain does not alter the shape or the homotetrameric oligomerization of Eg5, as was described previously (*Acar et al., 2013*; *Weinger et al., 2011*; *Figure 3B*; *Figure 3—figure supplement 1A–B*). Using TIRF microscopy, we reconstituted FL-Eg5-GFP and Eg5-Δtail-GFP motility along GMPCPP or Paclitaxel-stabilized AlexaF-633 labeled MTs (*Figure 3A*, lower panel). Both FL-Eg5-GFP and Eg5-Δtail-GFP exhibited robust and processive motility along MTs in 25 mMHEPES pH 7.5 containing 25 to 100 mM KCl (*Figure 3D–E*; *Video 2*). The processive motility we observe in pH 7.5 condition is highly homogeneous in contrast to the non-processive diffusive motility that has been seen at 80 mM PIPES pH 6.8 (BRB-80) with a range of 0–100 mM KCl conditions (*Kapitein et al., 2008*; *Kapitein et al., 2005*; data not shown). These observations, coupled with previous work on other kinesin motors, suggest that buffer composition, which influences pH and ionic strength, is critical for observing robust Eg5 motor motility along single MTs and affects landing rates along MTs (*Kapitein et al., 2008*).

We analyzed the motility properties of FL-Eg5-GFP and Eg5-Δtail-GFP motors at 25 mM HEPES pH 7.5 buffer conditions with added ionic strengths of 25, 50 and 100 mM KCl. At 25 mM KCl, FL-Eg5-GFP exhibited active yet extremely slow velocities (7 nm/s; *Table 4*; *Figure 3D*, left panel; *Video 3*), which was roughly four times lower than the velocity of Eg5-Δtail-GFP (31 nm/s; *Figure 3E*, left panel; *Video 2*). Quantitative analysis of motility events revealed mono-disperse distributions of velocities for the two motors that fit single Gaussians with single average values (*Figure 3F–G*, top panels). At 50 mM KCl, FL-Eg5-GFP exhibited a more than three-fold increase in average velocity (24 nm/s) compared to 25 mM KCl conditions (*Figure 3D*, middle panel). In contrast, Eg5-Δtail-GFP exhibited a nearly identical average velocity (35 nm/s) at 50 mM KCl, which is indistinguishable from what was observed at 25 mM KCl (*Table 4*; *Figure 3G*, middle panel). At 100 mM KCl, FL-Eg5-GFP motors exhibited two velocities (24 and 41 nm/s), with a bimodal distribution representing 60% and 40% of total, respectively (*Figure 3D* right panel; *Video 3*; *Figure 3F*, lower panel). Eg5-Δtail-GFP motors also exhibited two velocities at 100 mM KCl (36 nm/s and 55 nm/s) that were 20% higher than FL-Eg5-GFP and showed a similar bimodal distribution representing 85% and 15% of the total, respectively (*Table 4*; *Figure 3D* right panel; *Video 3*; *Figure 3G*, lower panel). At the 50 and 100 mM KCl conditions, fewer motor landing events were observed compared to 25 mM KCl (*Figure 3D–E*, right panels) for both FL-Eg5-GFP and Eg5-ΔTail-GFP. Thus, FL-Eg5 tetramers strongly respond to ionic strength, with 25 mM KCl significantly reducing its velocity, while Eg5-Δtail-GFP remains largely impervious to these effects. At 50 to 100 mM KCl, Eg5-Δtail-GFP consistently exhibited 20% higher velocities compared to FL-Eg5-GFP with bimodal distributions of slow and fast motors (*Table 4*).

We next explored the potential for FL-Eg5-GFP and Eg5-Δtail-GFP motile particles to form higher-order oligomers and sought to determine whether they form clusters based on the relative fluorescence intensities of motile GFP-fused motors (*Figure 3F–G*, inset graphs). Kymographs show that FL-Eg5-GFP motors in 25 mM KCl and Eg5-Δtail-GFP motors in 25 and 100 mM KCl conditions exhibited a homogeneous distribution of low-intensity values that was well fit by a single Gaussian (*Figure 3D–E*, left panels; *Video 3*; *Table 4*; *Figure 3F,G* insert graphs in top panels). In contrast, FL-Eg5-GFP particles exhibited both bright and dim intensity value distributions and a bimodal intensity value distribution at 100 mM KCl (*Figure 3D*, right panel; *Video 3*) which can be fitted by two overlapping Gaussians. FL-Eg5-GFP particles displayed high intensity values that were on average four-fold higher than the dim motile particles observed at 25 mM KCl, and low intensity values

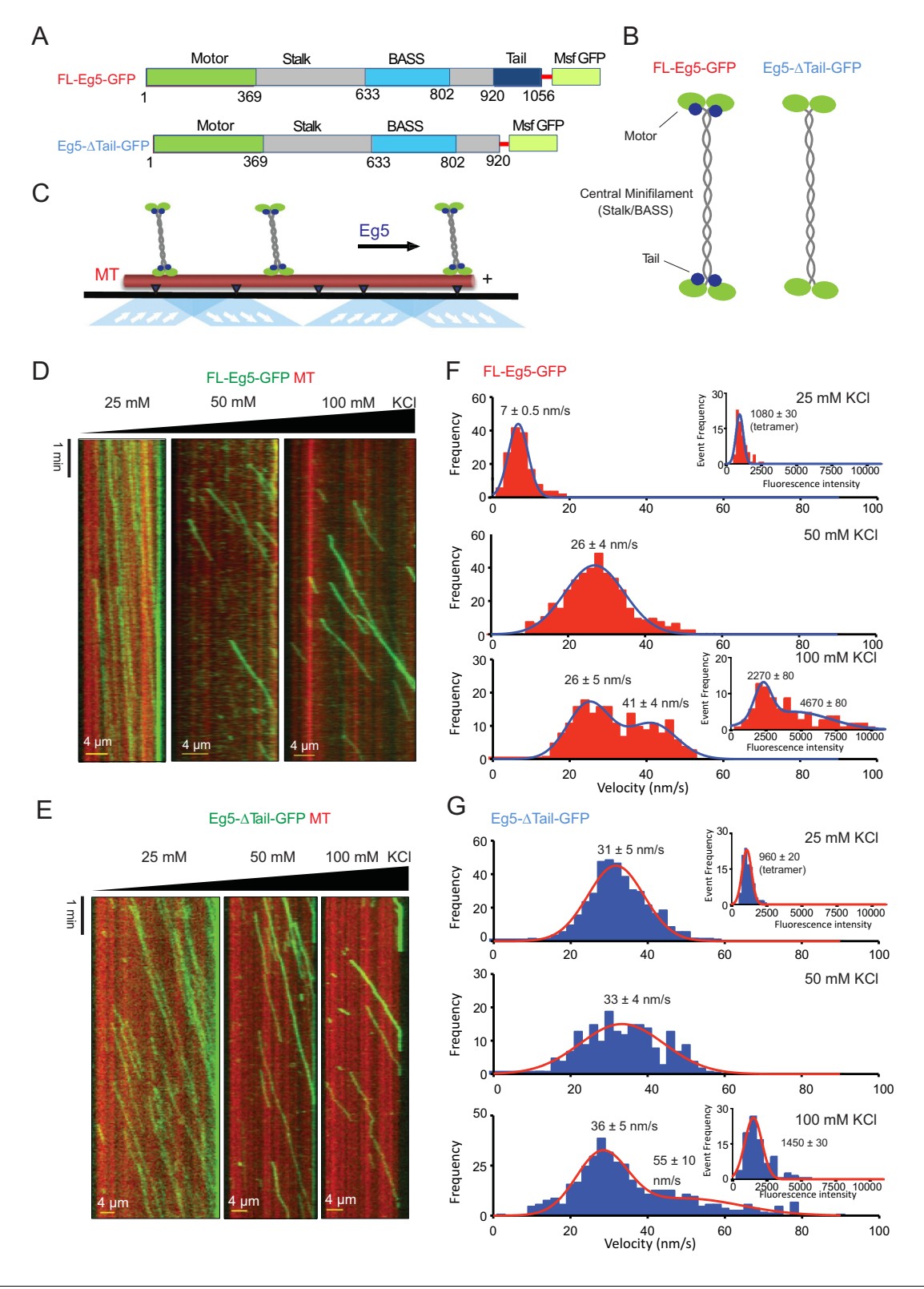

**Figure 3.** The kinesin-5 tail domain decreases velocity for homotetrameric motors along MTs. (**A**) Human Eg5 constructs used in reconstitution studies. Top panel, domain organization of FL-Eg5-GFP. Second panel, domain organization of Eg5-Δtail-GFP with its tail domain deleted (residues 920–1058) with the C-termini fused to monomeric superfolder GFP (msf-GFP). (**B**) The homotetrameric Eg5 organizations for the two constructs FL-Eg5-GFP and Eg5-Δtail-GFP, as described previously (***Acar et al., 2013***). (**C**) Scheme for TIRF microscopy of Eg5 motors (green) undergoing motility along single

*Figure 3 continued on next page*

*Figure 3 continued*

surface anchored AlexaF-633 and biotin labeled MTs via neutravidin-biotin attachment (red). (D) Kymographs of FL-Eg5-GFP motor motility along MTs in 25, 50 and 100 mM KCl pH 7.5 condition. Kymographs in dual color showing MTs (red) and GFP channels (green) are shown. Left panel, FL-Eg5-GFP motors undergo extremely slow motility at 25 mM KCl motor and their particle intensities are generally uniform. Middle panel, FL-Eg5 GFP undergoes motility with increased velocity and exhibits visual variation in particle intensity. Right panel, FL-Eg5-GFP motor show motility with higher velocity and exhibit bright and dim intensity particles at 100 mM KCl. Note FL-Eg5-GFP motors accumulate at MT plus-ends in 25 mM KCl; this plus-end residence decreases at 50 and 100 mM KCl. (E) Kymographs of Eg5-Δtail-GFP motility along anchored MTs at 25, 50 and 100 mM KCl pH 7.5 condition. Kymographs in dual color showing MTs (red) and GFP channels (green) are shown. Eg5-Δtail-GFP motors exhibit motility at similar velocities in all conditions. Note the homogeneity in motor intensities for Eg5-Δtail-GFP and its rapid motility at 25 mM KCl in contrast to the very slow motility of FL-Eg5-GFP. Motor intensities are uniform for Eg5-Δtail-GFP at 25 mM KCl and remain mostly homogeneously dim at 100 mM KCl. Note all Eg5-Δtail-GFP accumulate at MT plus-ends in a salt dependent manner. (F) Top panel, histogram for FL-Eg5-GFP motile particle velocity to frequency distribution reveals homogeneous and very slow velocity 25 mM KCl. Middle panel, histogram for FL-Eg5-GFP motor velocity to frequency distribution at 50 mM KCl reveals a 3-fold higher velocity than at 25 mM KCl. Bottom panel, histogram for velocity frequency bi-modal distribution for FL-Eg5-GFP at 100 mM KCl. Right inset panels show Eg5 fluorescence intensity distribution for motile particles at 25 and100 mM kCl. The fitted trend lines are shown in blue and averages are shown above each peak. These reveal that the motors are homogeneous homotetramers at 25 mM KCl, but cluster into dimers and tetramers of homotetramers at 100 mM KCl. Statistical t-tests are shown in *Figure 3—figure supplement 1F*. (G) Top panel, histogram for Eg5-Δtail-GFP motor motile particle velocity to frequency distribution (blue) at 25 mM KCl condition, revealing a three-folds faster than FL-Eg5-GFP at 25 mM KCl. Middle panel, histogram of velocity to frequency Eg5-Δtail-GFP motor particle distribution at 50 mM KCl revealing little change in velocity. Bottom panel, histogram of velocity to frequency distribution Eg5-Δtail-GFP motor particle revealing a bi-modal trend at 100 mM KCl. The fitted trend lines are shown in red and averages are shown above each peak. Inset panels at 25 and 100 mM KCl show that Eg5-Δtail-GFP motor fluorescence intensity distribution remain mostly as single homotetramers at 25 and 100 mM KCl. Statistical t-tests are shown in *Figure 3—figure supplement 1G*. The online version of this article includes the following figure supplement(s) for figure 3:

**Figure supplement 1.** The tail domain decreases the velocity of homotetrameric kinesin-5 motors along MTs.

that were roughly two-folds higher than the dim particles at 25 mM KCl (*Figure 3F*, insert graph in bottom panel; *Table 4*). Conversely, at 100 mM KCl, Eg5-Δtail-GFP motile particles exhibited a narrow intensity distribution that closely matched Eg5-Δtail-GFP motile particle distributions at 25 mM KCl (*Figure 3G*, insert graph in bottom panel; *Table 4*). Thus, FL-Eg5-GFP and Eg5-Δtail-GFP both show motility as homogeneous particles, likely individual homotetramers, with little clustering at 25 mM KCl. At 100 mM KCl, FL-Eg5-GFP motors form clusters consisting of up to two to four homotetramers at 100 mM KCl, unlike the Eg5-Δtail-GFP, which remained as individual homotetramers under all conditions tested. The tail domain may thus mediate inter-homotetramer interactions that induce clustering of Eg5 motors on the MT.

Analysis of the FL-Eg5-GFP and Eg5-Δtail-GFP motility run lengths revealed that the tail influences the processive motility of kinesin-5. At 25 mM KCl, both FL-Eg5-GFP and Eg5-Δtail-GFP motors exhibit run lengths that exceed the experimental imaging time with both motors eventually concentrating at MT plus-ends and were thus were not quantifiable (*Figure 3—figure supplement 1C*; *Table 4*). In contrast, at 50 mM KCl we observe measurable run lengths for FL-Eg5-GFP and Eg5-Δtail-GFP motility events that are identical and in the range of 13–14 μm. At 100 mM KCl, FL-Eg5-GFP retained long run lengths with an average of 13 μm, while Eg5-Δtail-GFP showed a 45% decrease in run lengths to an average of 8 μm (*Figure 3C*; *Table 4*). These data suggest that the tail plays a role in maintaining homotetramer-MT interactions, with deletion of the tail resulting in a less processive motility cycle and a decrease in its affinity to the MT lattice.

## The kinesin-5 tail domain is required for efficient zippering of anti-parallel MTs into sliding zones

To understand the impact of the tail-to-motor interaction on kinesin-5 MT motility under relative MT sliding conditions, we reconstituted Eg5 MT sliding *in vitro* using three-color TIRF assays in 25 mM HEPES, pH 7.5 with 25–50 mM KCl conditions (*Figure 4A*). AlexaF-633 and biotin-labeled MTs were attached to PEG-treated glass

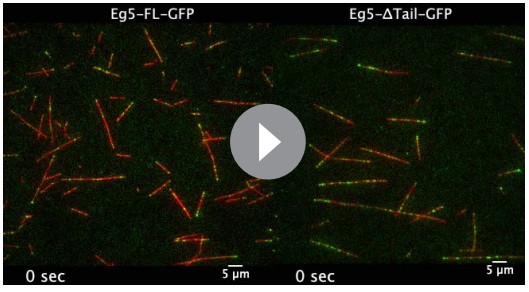

**Video 2.** Wide view of FL-Eg5-GFP and Eg5-Δtail-GFP (green) along single MTs (Red).
https://elifesciences.org/articles/51131#video2

**Table 4.** Motility parameters for FL-Eg5-GFP and Eg5-Δtail-GFP along single MTs.

| FL-Eg5-GFP | Motility (nm/s) | Motor Fluorescence (Au) | Run length (μm) |
| --- | --- | --- | --- |
| 25 mM KCl | 7 ± 0.5 n = 149 | 1080 ± 30 n = 100 | N/A |
| 50 mM KCl | 26 ± 4 n = 200 | N/A | 13.7 ± 0.6 |
| 100 mM KCl | 26 ± 5 (60%)<br>41 ± 4 (40%) n = 149 | 2277 ± 100<br>4467 ± 630 n = 92 | 13.08 ± 0.6 |
| Eg5-Δtail-GFP | | | |
| 25 mM KCl | 32 ± 5 n = 421 | 960 ± 20 n = 95 | N/A |
| 50 mM KCl | 33 ± 4 n = 420 | N/A | 14.9 ± 0.6 |
| 100 mM KCl | 36 ± 5 (85%)<br>55 ± 10 (15%) n = 149 | 1450 ± 3 n = 95 | 8.0 ± 0.6 |

surfaces via a neutravidin to biotin linkage (anchored MT, red). AlexaF-560 labeled MTs (free MT yellow) and FL-Eg5-GFP or Eg5-Δtail-GFP motors were then added to reconstitute the crosslinking and sliding of free-MTs (yellow) along the anchored MTs (red) (*Figure 4A*). Under these conditions we observed that 10–20 nM FL-Eg5-GFP motors promote robust MT sliding. FL-Eg5-GFP motors were able to bind along the anchored MTs, crosslink free-MTs (yellow), and zipper MTs to form anti-parallel MT sliding zones (*Figure 4A*-left; *Figure 4—figure supplement 1*; *Video 4*). Within a sliding event between two antiparallel MTs, FL-Eg5-GFP motors undergo slow motility similar to those observed along single MTs (*Figure 3*) and concentrate within newly formed two-MT bundles to produce sliding zones (green) (*Figure 4B*-left). To determine the effects of ionic strength on sliding, we measured motor and MT velocities at increasing solution ionic strength (25–50 mM KCl) (*Figure 4C*; *Table 5*). At 50 mM KCl, FL-Eg5-GFP motors displayed a two-fold increase in velocity similar to the effect along single MTs (*Figure 3*). This increase in motor velocity on anchored MTs matched a two-fold increase in the sliding rate of the free-MTs along the anchored MTs (*Figure 4C*; *Table 5*). Our experiments therefore show a correlation between the tail-mediated decrease in kinesin-5's velocity at lower ionic strength, and the velocity of the MTs in anti-parallel MT sliding (*Table 5*; *Figure 4C*).

We next studied the Eg5-Δtail-GFP in the MT sliding assays at similar motor concentrations (10–20 nM). Strikingly, the Eg5-Δtail-GFP motors crosslinked the free-MT only at focal points and were unable to completely zipper the free MT along the anchored MT (*Figure 4A*, right; *Figure 4B* right; *Video 4*). This MT zippering defect by the Eg5-Δtail-GFP motors lead the free MTs to 'scissor' along the anchored MTs due to Brownian motion at the crosslinking point (*Figure 4B*-right; *Video 4*). The scissoring behavior occurred most frequently near or at MT plus-ends, where the Eg5-Δtail-GFP

motors concentrate (*Figure 4B*, lower right, second and third panels). We next studied the effect of increasing Eg5-Δtail-GFP motor concentrations on the formation of MT sliding zones. A ten-fold higher concentration of Eg5-Δtail-GFP (200 nM) led to a higher number of MT sliding events relative to scissoring events (*Figure 4D*). At 20 nM Eg5-Δtail-GFP, only 6% of MT crosslinking events transitioned toward MT sliding events and the remaining 94% of events retained a scissoring defect. In contrast, at 200 nM Eg5-Δtail-GFP about 45% of crosslinking events transitioned into sliding events and 55% remained only scissoring events (*Figure 4D*). A possible reason for this is tail-mediated regulation leading to slower motility, which concentrates Eg5 motors in the overlap zone and facilitates MT zippering, as suggested by the fact that a high concentration of Eg5-Δtail-GFP did not fully

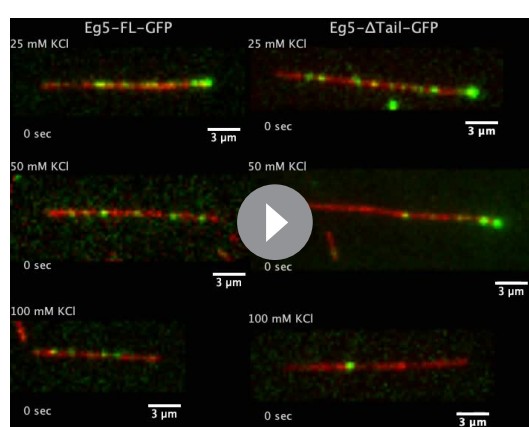

**Video 3.** close up views of FL-Eg5-GFP (left) and Eg5-Δtail-GFP (right) along single MTs at 25, 50 and 100 mM KCl conditions.
https://elifesciences.org/articles/51131#video3

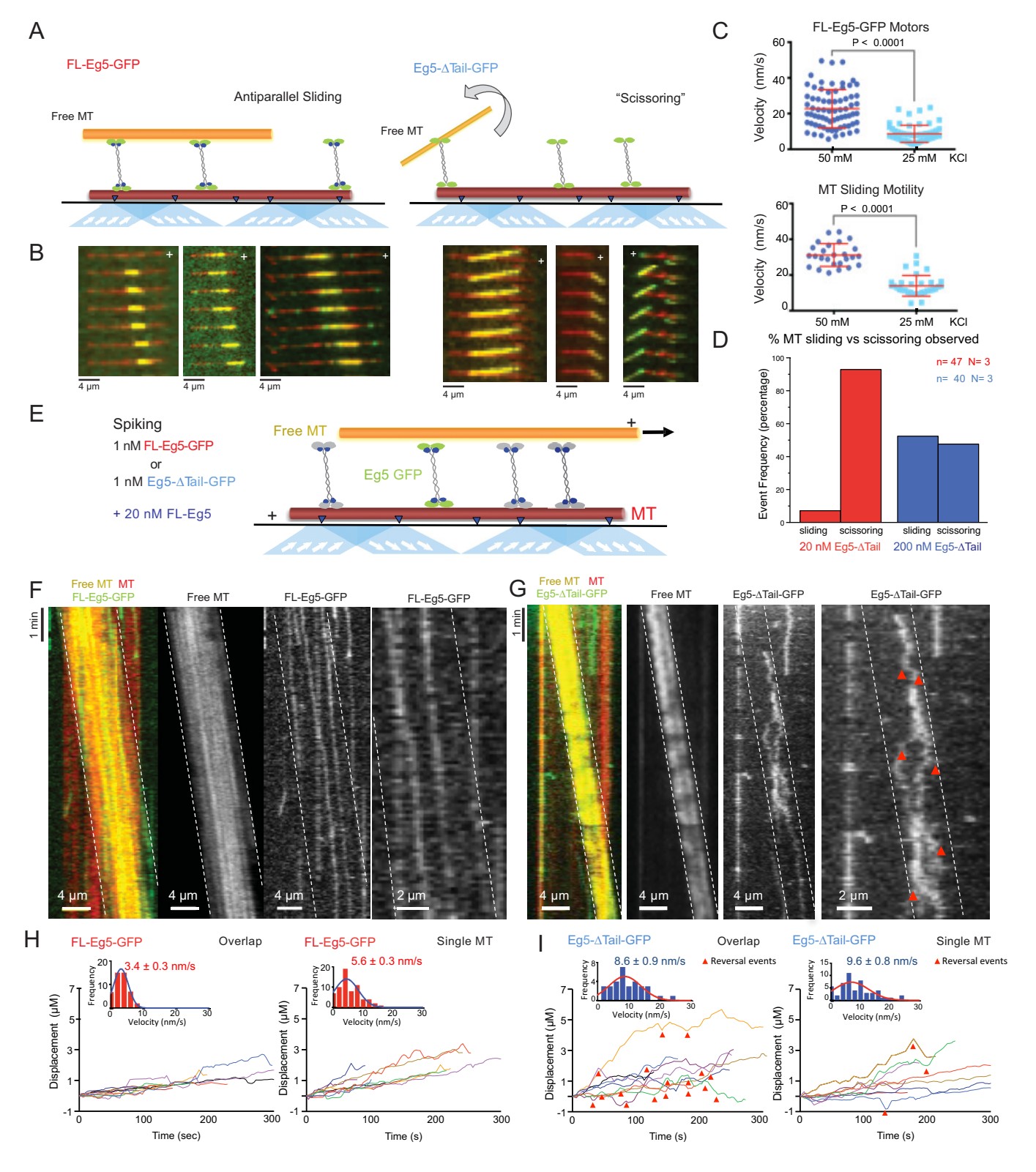

**Figure 4.** The kinesin-5 tail is critical for the zippering of two sliding MTs via slow directional motility within the overlap zones. (**A**) Left panel, scheme for TIRF microscopy reconstitution of MT sliding assays where FL-Eg5-GFP motors (green) recruit free MTs (orange) along single surface anchored AlexaF-633 and biotin labeled MTs via neutravidin-biotin (red). Right panel, scheme for TIRF microscopy MT sliding where Eg5-Δtail-GFP mediates crosslinking of the free-MT (orange) without zippering them along the AlexaF-633 and biotin labeled MTs via neutravidin-biotin (red). Their activity

*Figure 4 continued on next page*

*Figure 4 continued*

leads to scissoring motility. (B) Montages for two types of MT sliding events. Left panels, FL-Eg5-GFP mediates MT sliding. Right panels, Eg5-Δtail-GFP crosslinks MTs but does not zipper them along the anchored MT, leading to scissoring events. Additional examples in *Figure 4—figure supplement 1B* (C) The influence of ionic strength on FL-Eg5-GFP MT sliding activity. Top panel, FL-Eg5-GFP motor particle motility velocities (top panel) and free MT sliding velocity (lower panel). (D) The proportion of MT sliding and scissoring events in percentage of total observed for experiments at 20 and 200 nM Eg5-Δtail-GFP, respectively. The increase in Eg5-Δtail-GFP leads to higher proportion of MT sliding compared to scissoring events (N = 70–100 sliding/scissoring events for each condition). (E) Scheme for reconstitution of fluorescent Eg5 motor spiking in MT sliding assay with 20 nM FL-Eg5 (non-fluorescent, gray). 1 nM FL-Eg5-GFP or 1 nM Eg5-Δtail-GFP (green) were added to visualize single motors exhibit motility along the anchored MTs (red) and transition into the MT sliding zone with both free (yellow) and anchored MT (red). (F) Example kymographs of MT sliding spiking assays with 1 nM FL-Eg5-GFP in the presence of 20 nM FL-Eg5 (unlabeled). Three color kymographs are shown including the overlay (right panel), the free MT sliding (middle panel, yellow) being slide apart in the presence of single FL-Eg5-GFP motors (middle panel, green). Extreme right panels show a magnified views. Note the slow and directional motility of FL-Eg5-GFP motors. Additional examples in *Figure 4—figure supplement 1C*. (G) Example kymographs of MT sliding spiking assays with 1 nM Eg5-Δtail-GFP in the presence of 20 nM FL-Eg5 (unlabeled). Three color kymographs are shown including the overlay (right panel), the free MT sliding (middle panel, yellow) being slide apart in the presence of single Eg5-Δtail-GFP motors (middle panel, green). Extreme right panels show magnified view. Note FL-Eg5-GFP motors undergo bi-directional motility with stochastic switching (shown in red arrows) along either of the two anti-parallel MTs within sliding zones. Additional examples in *Figure 4—figure supplement 1C*. (H) Single motor track quantification of 1 nM FL-Eg5-GFP motor motility in the MT sliding zone (left, overlap) and along the anchored MT (right, single). Note FL-Eg5-GFP undergoes slow motility in general but its velocity decreases even further in the MT sliding zone. Average values reported in *Table 5* (I) Single motor track quantification of 1 nM Eg5-Δtail-GFP motor motility in the MT sliding zone (left, overlap zone) and along the single anchored MT (right, single-MT). Note Eg5-Δtail-GFP undergoes rapid motility in both zones, but its motility switches direction (reversals marked by red arrow heads) particularly within the overlap region of the MT sliding zone. Average values reported in *Table 5*.

The online version of this article includes the following figure supplement(s) for figure 4:

**Figure supplement 1.** The kinesin-5 tail domain regulates the zippering of two sliding MTs via slow directional motility within the overlapping zones.

---

restore the zippering of two MTs into sliding zones.

## The kinesin-5 tail domain is critical for slowing motility within MT sliding zones

To dissect the role of the tail-motor interface in kinesin-5 engagement within the MT sliding zones, we next studied how FL-Eg5-GFP or Eg5-Δtail-GFP motors behave outside and within active sliding zones formed by non-fluorescent FL-Eg5 (*Figure 4E*). FL-Eg5-GFP or Eg5-Δtail-GFP were spiked in at 1 nM into MT sliding assays containing 20 nM unlabeled FL-Eg5 to visualize single motors along the anchored MT and within active MT sliding zones (*Figure 4E*; see Materials and methods). FL-Eg5-GFP motors exhibit slow plus-end directed motility along anchored MTs with infrequent direction reversals, with their average velocity decreasing within MT sliding zones (5–7 nm/sec) (*Figure 4H*). In contrast, Eg5-Δtail-GFP motors exhibit an increased plus-end directed motility (10 nm/sec) along the anchored MT (*Figure 4I*). Upon entering MT sliding zones and interacting with both MTs, Eg5-Δtail-GFP motility remained rapid but the motors transitioned to bi-directional motility with frequent reversals (*Figure 4I*). We interpret these direction reversals to be uncoordinated MT plus-end directed motility along either of the two anti-parallel MTs within sliding zones that potentially indicate a higher rate of unbinding from the MT (*Figure 4G*, third panel from the left; *Video 5*). The Eg5-Δtail-GFP velocity is roughly two-fold higher than that of FL-Eg5 GFP along single MTs and within MT sliding zones. Eg5-Δtail-GFP also exhibited a three-fold shorter average run length compared to FL-Eg5-GFP within MT sliding zones (*Table 5*; *Figure 4F* right panels; *Figure 4I*; *Video 5*). The kinesin-5 tail thus is critical for Eg5 motors to engage within the MT sliding zones by modulating a unique motor-MT association which decreases motor unbinding rates, and potentially mediates coupling between the two MT-bound ends of kinesin-5. The bi-directional motility of Eg5-Δtail-GFP and its frequent direction reversals suggest that the two ends of the motor are undergoing rapid

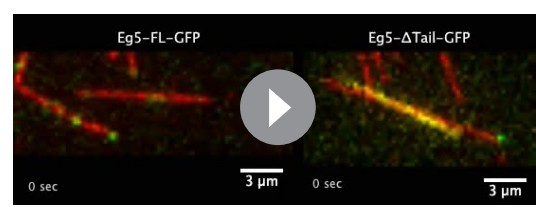

**Video 4.** left, close up views of FL-Eg5-GFP motors (green) mediating zippering of free MT (yellow) along anchored MT (red). Right close up view of Eg5-Δtail-GFP motors crosslinking but unable to zipper MTs leading to scissoring defect.
https://elifesciences.org/articles/51131#video4

**Table 5.** Motor motility and MT sliding parameters in vitro MT sliding assays.

**Single motor velocities in relation to free MT sliding motility**

| FL-Eg5-GFP | Free MT sliding motility (nm/s) | Motility in sliding zones (nm/s) |
| --- | --- | --- |
| 25 mM KCl | 13.8 ± 1.0 n = 26 | 13.9 ± 1.0 n = 32 |
| 50 mM KCl | 31.2 ± 1.2 n = 33 | 22.7 ± 1.2 n = 71 |
| **Single motor motility within MT sliding zones** | | |
| | Eg5-Δtail-GFP motors (nm/s) | FL-Eg5-GFP motors (nm/s) |
| Overlap Zone | 8.6 ± 0.9 n = 32 | 3.4 ± 0.3 n = 67 |
| Single MT | 9.6 ± 0.8 n = 52 | 5.6 ± 0.3 n = 45 |

uncoordinated ATP hydrolysis cycles as they bind both MTs within sliding zones.

## The tail domain is critical for kinesin-5 to generate high forces that slide apart MTs

In order to understand how the tail regulates the ability of kinesin-5 to slide apart antiparallel MTs, we next measured the forces generated by FL-Eg5-GFP or Eg5-Δtail-GFP on the free MT during sliding events using optical trapping combined with TIRF assays as previously described (*Shimamoto et al., 2015*; *Figure 5A*). Here, we reconstituted anti-parallel pairs of MTs similar to those described above (*Figure 4*), but with an additional step of attaching polystyrene beads coated with nucleotide-free kinesin-1 motor domain to the ends of Rhodamine-labeled MTs (red). These bead-associated free-MTs were reconstituted to form anti-parallel overlaps within flow chambers containing Hilyte-647-labeled MTs (purple) anchored to the slide surface and either FL-Eg5-GFP or Eg5-Δtail-GFP (green). These sliding MT pairs were then simultaneously imaged using TIRF microscopy while measuring the force exerted on the bead using an optical trap (*Figure 5A*). Under these conditions, 3–10 nM FL-Eg5-GFP produced robust MT sliding events (*Figure 5B* left panel; *Video 6*). In contrast, for 3–10 nM Eg5-Δtail-GFP, MT sliding was very rarely observed, and the majority of crosslinking events resulted in free MTs undergoing 'scissoring' movements about a single point on the anchored MT, similar to our previous observations in TIRF sliding assays (*Figure 4D*). In order to recruit comparable amounts of Eg5-Δtail-GFP motors within the overlap regions and to promote relative MT sliding, we increased the amount of Eg5-Δtail-GFP in the chamber by 50–100-fold (*Figure 4—figure supplement 1D*; *Figure 5B*; *Video 6*). Under these conditions, regions of MT overlap exhibited increased recruitment of kinesin-5 molecules (*Figure 5B*; *Video 6*).

To measure the pushing forces generated by MT sliding under these conditions, the Rhodamine-labeled MT-bound polystyrene bead was optically trapped upon observation of a sliding event (*Figure 5C*). FL-Eg5-GFP-mediated MT sliding exhibited a steady increase in the force produced until a maximum 'plateau' force was reached, frequently resulting in many 10 s of pN of force across the MT sliding pair. This behavior was quite similar in both timescale and force magnitude to reports of *Xenopus* Eg5 pushing forces measured in a similar assay (*Shimamoto et al., 2015*). In contrast, Eg5-Δtail-GFP-mediated MT pairs exhibited short excursions of force increase and reached significantly lower 'plateau' values with lower total forces generated overall (*Figure 5C*).

For each individual MT sliding event examined, we also calculated the integrated intensity of GFP signal within the overlap region as

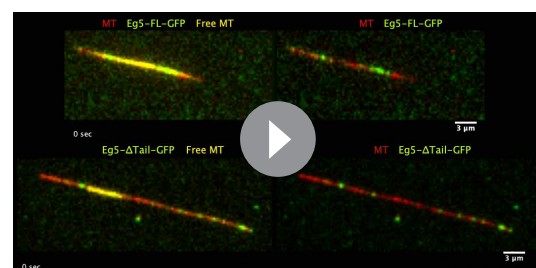

**Video 5.** Top left, close up view of 1 nM FL-Eg5-GFP motor (green) spiking during free MT (yellow) sliding along anchored MT (red) mediated by 20 nM unlabeled FL-Eg5. Top right, same event without the free MT revealing FL-Eg5-GFP motors along the anchored MT. Bottom left, close up view of 1 nM Eg5-Δtail-GFP motor (green) spiking during free MT (yellow) sliding along anchored MT (red) mediated by 20 nM unlabeled FL-Eg5. Bottom right, same event without the free MT revealing Eg5-Δtail-GFP motors along anchored MT.
https://elifesciences.org/articles/51131#video5

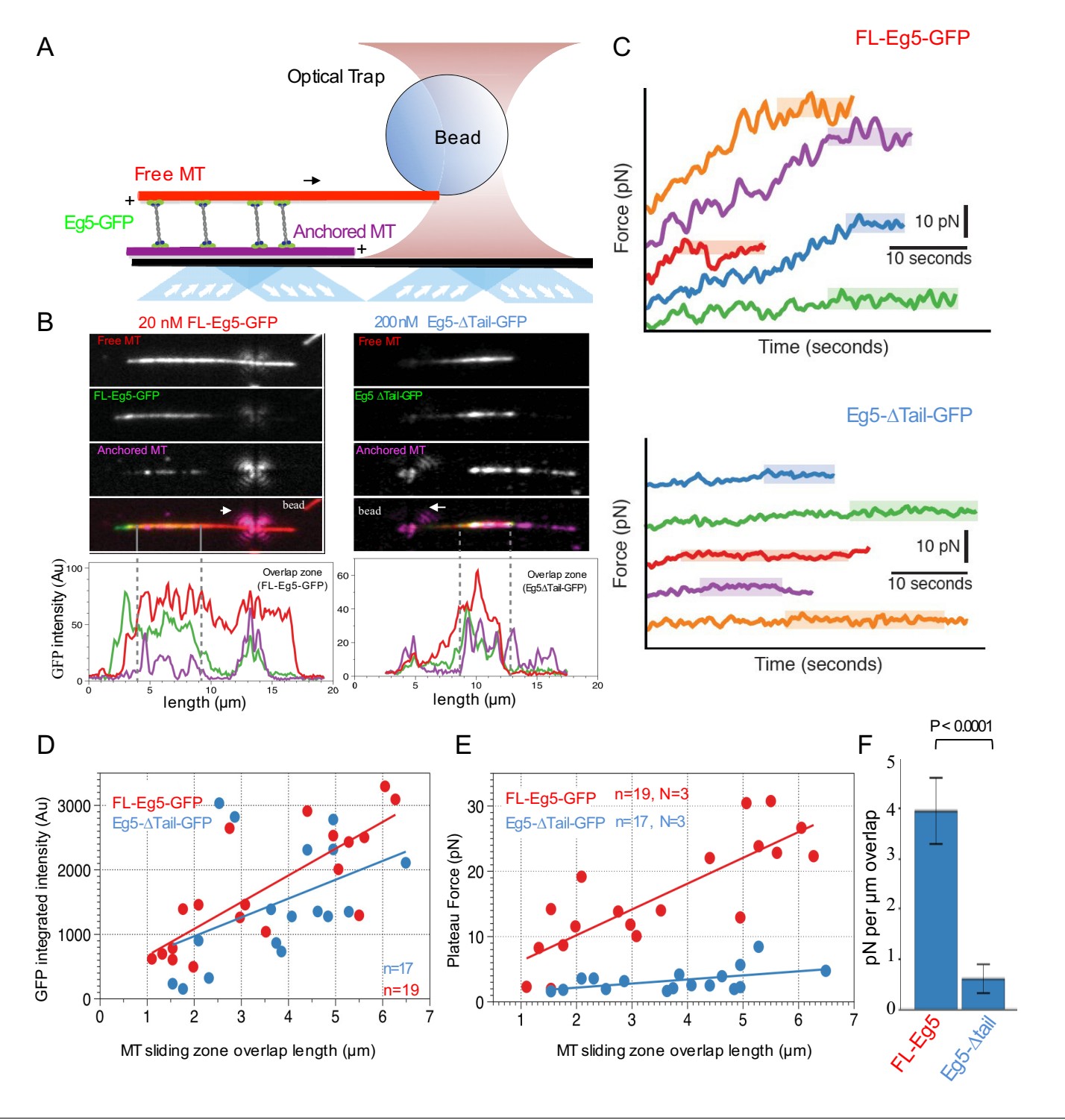

**Figure 5.** Kinesin-5 tail is critical in generating pushing forces during MT sliding. (**A**) Scheme for MT sliding and optical trapping to measure the MT sliding pushing forces. Hilyte 649 labeled and biotins labeled MTs (purple) were attached to glass surfaces via neutravidin-biotin. FL-Eg5-GFP motors (green) slide apart the Rhodamine-labeled MTs (red) while bound to a polystyrene bead (sphere) coated with kinesin-1 rigor mutant protein, which becomes locked into the optical trap to measure forces. (**B**) Example images of MT sliding events mediated by FL-Eg5-GFP (left) and Eg5-Δtail-GFP (right). Top, free MT shown in gray scale. Second, Eg5-GFP intensity in the overlap zone. Third, attached MT. The polystyrene bead can be seen attached to the free MT. Lower panel, fluorescence intensity for each of the channels above showing the identification of the overlap zone length and total GFP intensity above background. (**C**) Example optical trapping force profiles generated for sliding events by FL-Eg5-GFP (top) and Eg5-Δtail-GFP

*Figure 5 continued on next page*

*Figure 5 continued*

(bottom). Top panel, FL-Eg5-GFP MT sliding events generate and build up forces that then plateau (highlighted level). Lower panel, Eg5-Δtail-GFP MT sliding events generate very weak forces, which plateau at lower values. (D) Scaled comparison for overlap zone length (μm) plotted in relation to the overall GFP intensity for each sliding event. FL-Eg5-GFP is shown in red while Eg5-Δtail-GFP is shown in blue. Note there is generally little discernible statistical difference between the slopes of these comparisons. (E) Scaled comparison of the plateau forces generated (pN) in relation to the size of the overlap zone for each MT sliding event. FL-Eg5-GFP is shown in red while Eg5-Δtail-GFP is shown in blue. Note the slope of the FL-Eg5-GFP force to length comparison is steeper than the Eg5-Δtail-GFP force to length comparison (F) Forces generated per μm of overlap length representing the slope of linear comparison in E.

defined by the distance between MT plus-ends (dashed gray lines, *Figure 5B*). This served as a proportional readout of the amount of motors localized within the overlap region. By comparing a number of MT sliding zones, we determined there is an approximately linear relationship between the relative MT overlap zone length and the number of motors within this region (*Figure 5D*). Furthermore, a comparable concentration of FL-Eg5-GFP and Eg5-Δtail-GFP were measured within overlap zones, suggesting that similar numbers of motors were engaged within overlaps of similar size, despite the different concentration used in the assays (*Figure 5D*).

We also examined the relationship between the magnitude of the force plateau reached in each individual MT sliding event and the length of the MT overlap zone during force generation. For both Eg5-GFP and Eg5-Δtail-GFP we observed a nearly linear increase in plateau force relative to MT overlap zone length (*Figure 5E*). However, the slopes of these relationships differed significantly between the two constructs. Here the slopes represent the plateau pushing force generated per μm length of sliding zone: the Eg5-Δtail-GFP generates about 0.6 ± 0.3 pN per μm while the FL-Eg5-GFP generates about 4.0 ± 0.6 pN per μm, indicating about a seven-fold difference in MT sliding forces (*Figure 5F*). The pushing forces produced within sliding anti-parallel MT bundles by ensembles of Eg5-Δtail-GFP are roughly seven-fold lower than that those generated by comparable numbers of FL-Eg5-GFP motors. Together, these data indicate that the tail domain is critical for homotetrameric kinesin-5 motor sliding behavior and mediates regulation of force production during MT sliding events.

## The tail domain is critical for kinesin-5 localization to metaphase and anaphase mitotic spindles *in vivo*

We next determined the role of the kinesin-5 tail domain in motor localization of Eg5 in metaphase and anaphase in HeLa cells. Cells were transfected with human GFP-α-tubulin to visualize MTs, and either FL-Eg5-mCherry (FL-Eg5-mCh), Eg5-Δtail-mCherry (Eg5-Δtail-mCh), or mCherry alone (mCh). Expression of each construct in HeLa cells was assessed by western blot (*Figure 6A*). In fixed metaphase cells, FL-Eg5-mCh localized to spindle MTs. In contrast, localization of Eg5-Δtail-mCh was more diffuse, with increased cytoplasmic signal (*Figure 6B*). This difference in localization was quantified as the ratio of mCh signal localized to the mitotic spindle and mCh signal in the cytoplasm. In cells with comparable mCh-construct expression levels (*Figure 6C*, left), this spindle-to-cytoplasm ratio was high for FL-Eg5-mCh (2.39 ± 0.08, mean ± SEM) and significantly reduced for Eg5-Δtail-mCh (1.37 ± 0.02, p<0.0001, ANOVA with Tukey's multiple comparisons test), indicating that deletion of the tail causes a defect in localization to metaphase spindle MTs (*Figure 6C*, right). Treatment with the compound BRD-9876, which locks Eg5 in a nucleotide-free-like state (*Chen et al., 2017*), only partially rescued the localization of Eg5-Δtail-mCh to metaphase spindle MTs, potentially due to defects in both the on and off-rates (*Figure 6D and E*).

Live cell imaging was used to assess whether this tail deletion localization defect was present in anaphase as well as metaphase mitotic spindles. Measurement of spindle to cytoplasm mCh

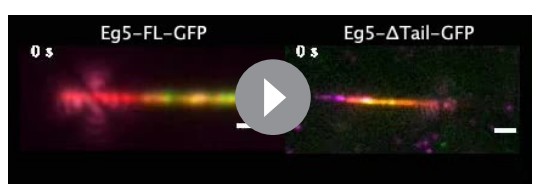

**Video 6.** left, optical trapping of MT sliding experiments revealing the bead attached to sliding free MT (red) along anchored MT (purple) mediated by FL-Eg5-GFP motors. Right, optical trapping of MT sliding experiments revealing the bead attached to sliding free MT (red) along anchored MT (purple) mediated by Eg5-Δtail-GFP motors.
https://elifesciences.org/articles/51131#video6

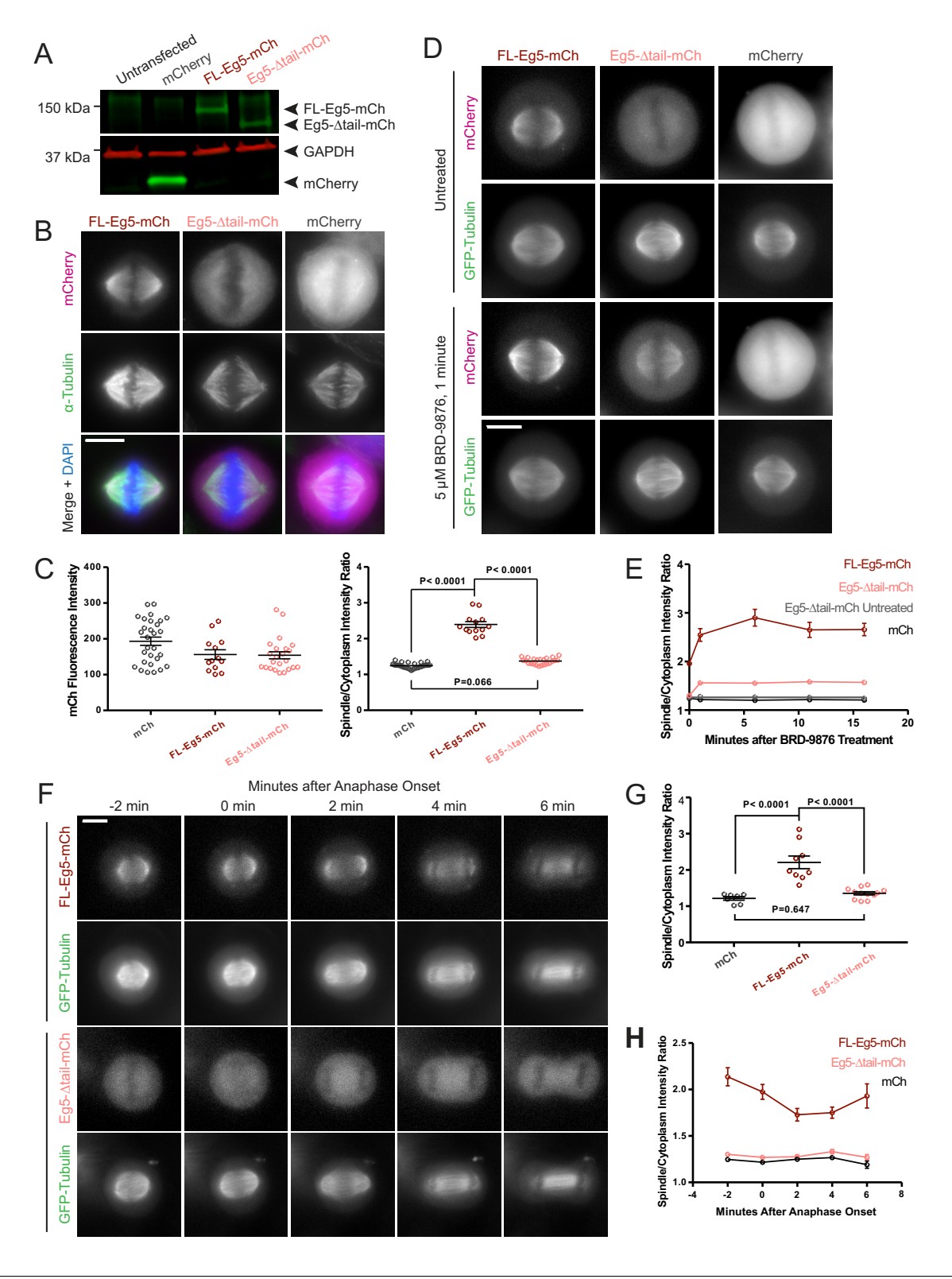

**Figure 6.** Deletion of the kinesin-5 tail domain disrupts localization of the motor to the mitotic spindle in metaphase and anaphase. (**A**) Western blot for mCherry (mCh, green) and GAPDH (red) indicating the expression of FL-Eg5-mCherry (FL Eg5-mCh, green) and Eg5-Δtail-mCherry (Eg5-Δtail-mCh, green) in HeLa cells. Results are representative of three independent experiments. (**B**) Localization of mCh, FL-Eg5-mCh, and Eg5-Δtail-mCh in HeLa cells arrested in metaphase via treatment with MG-132. FL-Eg5-mCh localized to spindle MTs. Tail deletion disrupted localization, and Eg5-Δtail-mCh

*Figure 6 continued on next page*

*Figure 6 continued*

signal was distributed between spindle MTs and the cytoplasm. Scale bar 10 µm. Images are representative of three independent experiments. (**C**) Left panel, mCh fluorescence intensities of single cells used for quantification of localization to spindle MTs (n = 13–29 cells per transfection condition, three independent experiments). Right panel, the ratio of mCh fluorescence signal on the spindle to signal in the cytoplasm was significantly lower in fixed metaphase cells expressing Eg5-Δtail-mCh compared to FL-Eg5-mCh, indicating reduced localization of Eg5-Δtail-mCh to spindle MTs (n = 13–29 cells per transfection condition, three independent experiments, p values from ANOVA with Tukey's post hoc test). (**D**) Treatment of live HeLa cells expressing Eg5-mCh constructs and GFP-Tubulin with the Eg5 rigor inhibitor BRD-9876 resulted in a rapid (<1 min) increase in FL-Eg5-mCh signal on the spindle. Inhibitor treatment increased, but did not fully rescue, localization of Eg5-Δtail-mCh to the spindle. Scale bar 10 µm. Images are representative of three independent experiments. (**E**) The ratio of mCh fluorescence signal on the spindle to signal in the cytoplasm rapidly increased after treatment with BRD-9876 in cells expressing FL-Eg5-mCh or Eg5-Δtail-mCh. The spindle-to-cytoplasm intensity ratio of Eg5-Δtail-mCh expressing cells never reached that of cells expressing FL-Eg5-mCh, indicating only partial rescue of motor localization with rigor inhibitor treatment. BRD-9876 treatment did not alter the ratio of mCh control cells (n = 7–13 cells per transfection condition, three independent experiments). (**F**) Deletion of the tail domain disrupted localization of Eg5 to the spindle in anaphase. Paired rows of images demonstrate the localization of FL-Eg5-mCh and Eg5-Δtail-mCh as HeLa cells expressing GFP-tubulin transitioned from metaphase to anaphase. FL-Eg5-mCh signal was observed at the spindle throughout the metaphase to anaphase transition and the motor localized to the midzone after anaphase onset (see 4–6 min panels). Increased cytoplasmic and reduced spindle signal was observed in cells expressing Eg5-Δtail-mCh throughout the metaphase to anaphase transition. Scale bar 10 µm. Images are representative of three independent experiments. (**G**) The ratio of mCh fluorescence signal on the spindle to signal in the cytoplasm was measured six minutes after anaphase onset. As in metaphase cells, localization of Eg5-Δtail-mCh to the spindle was significantly reduced compared to FL-Eg5-mCh (n = 7–12 cells per transfection condition, three independent experiments, p values from ANOVA with Tukey's post hoc test). (**H**) The spindle-to-cytoplasm intensity ratio of cells expressing Eg5-Δtail-mCh was lower than that of cells expressing FL-Eg5-mCh throughout the metaphase to anaphase transition, indicating a persistent localization defect caused by deletion of the tail domain (n = 7–12 cells per transfection condition, three independent experiments).

intensity ratios in anaphase showed a similar reduction, indicating decreased localization to spindle MTs, for Eg5-Δtail-mCh compared to FL-Eg5-mCh as seen in metaphase cells (ratios 1.36 ± 0.04 and 2.21 ± 0.17, mean ± SEM, respectively, p<0.0001, ANOVA with Tukey's multiple comparisons test) (***Figure 6F and G***). As HeLa cells progressed from metaphase to anaphase, the deletion of the tail consistently reduced the localization of Eg5 to spindle MTs (***Figure 6F and H***). These data support a critical role for the tail domain in kinesin-5 mitotic spindle MT localization.

## Discussion

Kinesin-5 motors share a conserved anti-parallel MT sliding activity, which is essential for the assembly and elongation of mitotic spindles. This conserved activity allows kinesin-5 motors to mobilize each of their MT tracks simultaneously as cargos for transport. Here, we show that a conserved tail-to-motor interaction at the two ends of the kinesin-5 homotetramer is responsible for kinesin-5 conserved MT sliding function (***Figure 7A–B***). The regulation is most clearly observed at low ionic strength conditions, where the tail exhibits the highest affinity to the motor domain and the highest level of MT-stimulated ATP hydrolysis regulation leading to a longer lived and highaffinity binding on MT lattices during its catalysis cycle (***Figure 1***; ***Figure 7B***). Structurally, the tail domain binds the motor domain in either the nucleotide-free or ADP MT-bound states by engaging the N-terminal subdomain at the α0-helix hairpin (***Figure 7C***). This element of the kinesin-5 motor domain rotates upward from the MT lattice in the nucleotide free state relative to a downward positioning upon binding nucleotide in the ATP-like state. The tail-induced stabilization of the upward state likely slows ATP binding into the active site of each motor during the ATP catalysis cycle (***Figure 2***; ***Figure 7C***). We observed processive FL-Eg5-GFP homotetramer motility along individual MTs, a behavior previously not observed likely due to the lower than physiological pH (pH 6.8) conditions used in previous studies (***Kapitein et al., 2008***). The tail-motor interaction decreases the MT-stimulated ATP hydrolysis rate resulting in very slow motility for FL-Eg5-GFP at 25 mM KCl along individual MTs and within MT sliding zones (***Figures 3–4***). Increasing the solution ionic strength (50–100 mM KCl) weakens tail-motor interface partially and relieves the suppression of ATP hydrolysis, leading to increased motility velocities. The tail domain also enhances the clustering of kinesin-5 homotetramers into oligomeric assemblies. At higher ionic strength (50 mM KCl), the MT sliding velocity directly correlated with an increased velocity of FL-Eg5-GFP motors within MT sliding zones. In contrast, Eg5-Δtail-GFP motors exhibited no suppression in motor velocity at 25 mM KCl, maintained a mostly constant average velocity at higher ionic strengths (50–100 mM KCl) and exhibited no

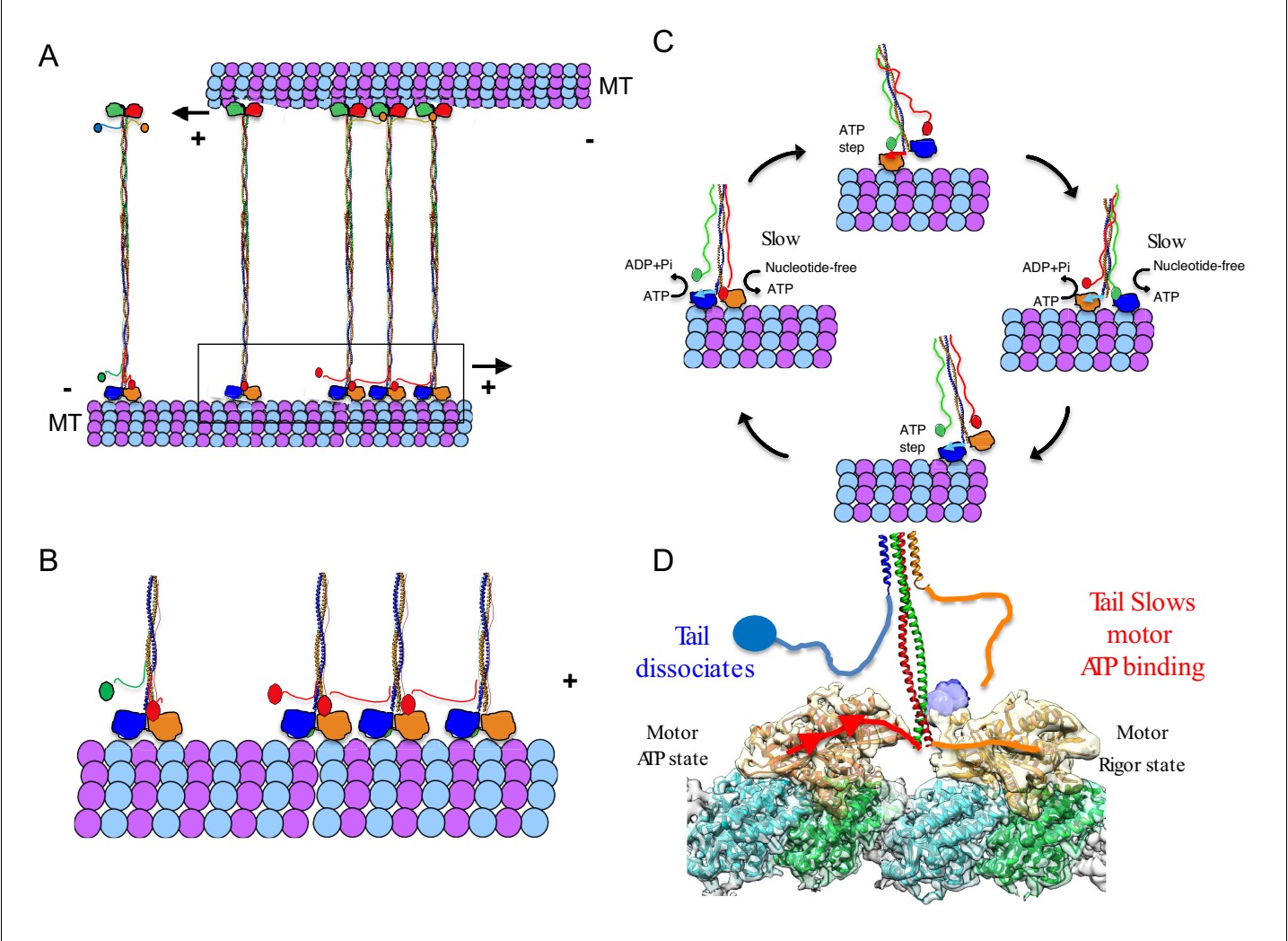

**Figure 7.** A revised model for kinesin-5 tail-motor interaction during stepping motility and critical role for force generation during MT sliding. (**A**) Model for kinesin-5 homotetramers with their motor and tail domains at each end of the bipolar minifilament (60 nm). The motors and tail domains at each end may form assemblies where the tail domain of one homo-tetramer makes contact with motors of a second tetramer to form clusters of two to four motors (**B**) The role of the tail in regulating the kinesin-5 hand-over-hand motility cycle by slowing ATP binding of the lead motor leading to slow hand over-hand motility and prevalence of the dual bound state at each end of kinesin-5. (**C**) The conformations of the kinesin-5 motor subdomains in the process of ATP binding. Left, The N-terminal subdomain (blue) is in upward state with the helix-0 to engage the tail domain and wedging the nucleotide binding pocket open. Right, the N-terminal subdomain moves downward enclosing on the bound ATP, leading the helix-0 to move downward and disengage from the tail domain. (**D**) Synthesized view of dual dimeric motor bound state of the kinesin-5 motor end. This state was synthesized based on the cryo-EM maps and, in vitro reconstitution, biochemical and kinetic studies described here. The tail makes contact with the motor domain only in the nucleotide-free state but dissociates when the motor is in the ATP state. The lead motor is bound to the tail while the trailing motor dissociates from the tail domain.

clustering behavior. The Eg5-Δtail-GFP velocity also remained consistently 20% higher than that of Eg5-FL-GFP at 50 and 100 mM KCl, suggesting that some tail regulation remains in place at even higher ionic strength. The weakened tail-to-motor interaction may play a role in enhancing the intermolecular interactions between kinesin-5 homotetramers as it resulted in higher-order clustering among FL-Eg5-GFP tetramers at 100 mM KCl, (*Figure 7A*, broken lines). The tail domain also enhances processive motility run lengths for FL-Eg5-GFP at higher ionic strengths, which decreased by 45% for Eg5-Δtail-GFP (*Figure 3—figure supplement 1B*). These data are fairly consistent with previous observations for full-length and tail-deleted *Xenopus* Eg5 (*Weinger et al., 2011*).

Our studies suggest that the tail-to-motor interaction of kinesin-5 is critical for coordinating the motility activities that are required in promoting MT sliding motility. We show that human FL-Eg5-

GFP motors capture antiparallel MTs and zipper them into sliding MT bundles (*Figure 4*). Loss of the tail-to-motor interaction in Eg5-Δtail-GFP motors does not prevent initial crosslinking of MTs at or near MT-plus-ends, but leads to a severe defect in the zippering of two MTs into sliding bundles (*Figure 4*). Within active MT sliding zones, we observe that the Eg5-Δtail-GFP motors are persistently bi-directional and moving along either antiparallel MTs likely due to loss of coordination between the two bipolar kinesin-5 motile ends (*Figure 4F*, left panel). In contrast, FL-Eg5-GFP undergoes smooth, slow and unidirectional motility towards the MT plus-end of the anchored MT. FL-Eg5-GFP motors generate seven-fold higher pushing forces compared to Eg5-Δtail-GFP motors while sliding apart antiparallel MTs. The forces generated by FL-Eg5-GFP likely result from the kinesin-5 tail-to-motor interface modulating motility along each MT within the sliding zone. The tail enhances the FL-Eg5-GFP motor-MT affinity leading to tightly engaged and slow-moving motors on both MTs within the sliding zone.

## A revised model for kinesin-5 sliding motility: Tail domains are essential for facilitating slow motility and high force production between two sliding MTs

Our studies suggest that the tail domain regulates processive hand-over-hand stepping during kinesin-5 motility (*Figure 7B–D*; *Vale and Milligan, 2000*). Well-studied MT plus-end-directed kinesin motors (i.e. kinesin-1) undergo hand-over-hand walking motility along MT protofilaments utilizing dimeric motor domains coupled by their neck linkers and neck coil-coils. During hand-over-hand motility, one motor domain of a dimeric kinesin binds the MT lattice in a trailing position, while the other motor domain occupies a leading position, 8 nm apart along two consecutive αβ-tubulins within MT protofilaments (*Figure 7D*). The trailing motor is in an ATP-like state with its neck-linker docked, while the lead motor initially binds weakly, leading to ADP dissociation and a nucleotide-free state (*Figure 7D*). The hand over hand cycle of dimeric kinesin-1 motors result in 8 nm steps toward the MT plus-ends (*Vale and Milligan, 2000*). Our studies suggest a new form of regulation for kinesin-5 hand-over-hand motility, where the tail domain decreases ATP binding and stabilizes the ADP or nucleotide-free states of the lead motor domain (*Figure 7C–D*). The kinesin-5 tail binds the α0-helix within the motor domain, stabilizing its N-terminal subdomain in an upward conformation and resulting in an open active site that slows its ATP binding capacity (*Figures 1–3*; *Figure 7C–D*). Our model is the simplest explanation for all the data presented here. The tail-motor regulatory mechanism, described here, is likely to be the source of kinesin-5's conserved sliding force-generating capabilities and is essential for its anti-parallel MT sliding activity. In yeast kinesin-5 motors, such as Cin8, these interactions may also regulate the reversal of direction from minus-end-directed motility along single MTs to plus-end-directed motility within sliding zones as tail deletion in Cin8 interferes with directionality reversal (*Düselder et al., 2015*).

The tail-motor interface may stiffen both kinesin-5-MT interactions at both ends of the homotetrameric filament within MT sliding zones and thus may improve force transmission between the two bipolar ends (*Figures 5* and *7B–D*). Our data suggests that the tail-motor interaction increases the time each end of the kinesin-5 homotetramer spends in the dual motor-bound state. The tail may promote a high affinity state for the motor domains at each of the homotetramer (*Figure 7C–D*). In the absence of the tail domain, Eg5-Δtail-GFP motors are unable to effectively engage both sliding MTs and exhibit reversal in motility direction by moving to the MT plus-end on either sliding MTs asynchronously. This leads to a poor capacity for generating MT sliding and pushing forces (*Figures 4–5*). Our data also indicate that this interaction may be responsible for clustering multiple homotetramers into larger complexes under certain conditions (*Figure 7A*). Clusters of up to four kinesin-5 homotetramers were observed in the yeast ortholog, Cin8, where cluster formation induced motility direction reversal and generated sites for the capture of free MTs to promote MT sliding (*Shapira et al., 2017*). We suggest that the clustering of kinesin-5 motors may serve to coordinate the motor motility cycles of groups of motors within MT sliding zones (*Figure 7A*). Cooperativity may also be a result of multiple motors stalled in a traffic jam due to slow stepping, a model previously suggested for Cut7, the *S. pombe* ortholog (*Britto et al., 2016*).

## Tail regulation of the kinesin-5 motor domain maybe modulated by mitotic kinases and the tail is critical for mitotic kinesin-5 functions

The kinesin-5 tail domain is essential for mitotic spindle assembly and elongation functions. Our studies confirm that the mitotic localization defect in Eg5-Δtail-GFP may relate to its rapid binding and unbinding from spindle MTs; trapping Eg5-Δtail-GFP motor in the nucleotide-free state using chemical strategies dramatically enhances their MT spindle localization (*Figure 6*). A recent study identified Eg5 mutations in cultured cells that result in resistance to Eg5 inhibitors (*Sturgill et al., 2016*). Among these is a mutant cell strain that possesses an Eg5 mutant with motor domain mutation that traps it in the nucleotide-free state that forms a near-normal bipolar metaphase spindle and for which correct mitotic organization is restored in the presence of specific compounds (*Sturgill et al., 2016*). This data highlights the importance of the stabilization of the nucleotide-free state during the force generation cycle of kinesin-5.

The tail-motor interface is essential for the stable localization of kinesin-5 motors to mitotic spindle MTs by also acting as a regulatory site for phosphorylation. The tail domain contains the conserved BimC box, a mitotic CyclinB dependent kinase phosphorylation consensus site (*Blangy et al., 1995*; *Sharp et al., 1999*). This motif is conserved across kinesin-5 motors and its mitotic phosphorylation was shown to promote the accumulation of kinesin-5 motors at the mitotic spindle midzone and to mediate spindle elongation in anaphase (*Sharp et al., 1999*). It remains unclear how phosphorylation of the BimC box influences the kinesin-5 tail-motor interaction. It is possible that phosphorylation enhances motor regulation by the tail and further slows MT-based motility to increase MT sliding efficiency during anaphase. Future studies of kinesin-5 motors phosphorylated at the BimC box will provide further clues for kinesin-5 regulation throughout the cell cycle.

### Concluding remarks

We present a new mechanism for the regulation of the kinesin-5 motor via its tail domain during plus-end-directed motility along MTs. The tail binds the motor domain to stabilize it in the high affinity MT-bound nucleotide-free state. This regulatory role is critical for slowing kinesin-5 motility along MTs allowing motors to crosslink and generate force to slide apart antiparallel MTs, an aspect of kinesin-5 that is essential for cell division to occur.

# Materials and methods

**Key resources table**

| Reagent type (species) or resource | Designation | Source or reference | Identifier | Additional Information |
|---|---|---|---|---|
| Chemical compound, drug | ATP | Sigma | A-2383 | *Figures 1*, *3*, *4*, *5* and *6* |
| Chemical compound, drug | ADP | Sigma | A-2754 | *Figure 1* |
| Chemical compound, drug | GTP | Sigma | G-8877 | *Figures 1*, *3*, *4* and *5* |
| Chemical compound, drug | GMPCPP | Jenna Biosciences | NU-405L | *Figures 3*, *4* and *5* |
| Chemical compound, drug | AMPPNP | Sigma | A-2647 | *Figure 1* |
| Chemical compound, drug | Paclitaxel | Sigma | T7402 | *Figure 1*, *2* |
| Other | Streptactin XT | IBA-life sciences | 2-1003-100 | *Figure 1*, *3* |
| Chemical compound, drug | d-Biotin | Sigma | B-4501 | *Figure 1*, *3* |
| Commercial assay or kit | EnzCheck ATPase assay kit | Thermofisher | E6646 | *Figure 1* |
| Chemical compound, drug | NeutrAvidin | Thermofisher | 31000 | *Figure 3*, *4* |

*Continued on next page*

Continued

| Reagent type (species) or resource | Designation | Source or reference | Identifier | Additional Information |
|---|---|---|---|---|
| Chemical compound, drug | biotin-PEG-3400-silane | Laysan Bio | Biotin-PEG-SIL-3400–500 mg | *Figure 3*, *4* |
| Chemical compound, drug | PEG-2000-silane | Laysan Bio | MPEG-SIL-2000–1 g | *Figure 3*, *4* |
| Chemical compound, drug | Pluronic-F127 | Sigma | P2443 | *Figure 3*, *4* |
| Antibody | anti-GAPDH (mouse monoclonal) | Thermo-Fisher | 437000 | Western blot: 1:10,000 |
| Antibody | anti-mCherry | Abcam | ab167453 | Western blot: 1:1000 |
| Antibody | anti-mouse IRDye680 (goat polyclonal) | LI-COR | 92568070 | Western blot: 1:10,000 |
| Antibody | anti-rabbit IRDye800 (goat polyclonal) | LI-COR | 92632211 | Western blot: 1:10,000 |
| Antibody | anti-tubulin DM1α (mouse monoclonal) | Sigma | T9026 | Immuno fluorescence: 1:750 |
| Antibody | anti-mouse AlexaFluor 488 (goat polyclonal) | Invitrogen | A-11029 | Immuno fluorescence: 1:500 |
| Antibody | anti-mouse AlexaFluor 647 (goat polyclonal) | Invitrogen | A-21236 | Immuno fluorescence: 1:500 |
| Commercial assay or kit | Nucleofector Cell Line SE Kit | Lonza | V4XC-1024 | |
| Commercial assay or kit | Phusion Site-Directed Mutagenesis | Thermo Scientific | F541 | *Figure 6* |
| Chemical compound, drug | BRD-9876 | Tocris Bioscience | 5454/50 | *Figure 6* |
| Chemical compound, drug | MG-132 | Selleckchem | S2619 | *Figure 6* |
| Peptide, recombinant protein | *Drosophila KLP61F* | UniprotKB/Swiss-Prot | P46863 | |
| Peptide, recombinant protein | *Human Eg5 (KIF11)* | UNiportKB/Swiss-Prot | P52732 | |
| Peptide, recombinant protein | *Porcine alpha tubulin* | UniprotKB/Swiss-Prot | Q2XVP4 | |
| Peptide, recombinant protein | *Porcine beta-tubulin* | UniprotKB/Swiss-Prot | P02550 | |
| Cell line (*E. coli*) | SoluBL21 bacterial expression | AmsBio | C700200 | *Figure 1*, *2* |
| Cell line (*S. frugiperda*) | Spodoptera frugiperda-9 (Sf-9 cells) | Thermofisher | 11496–015 | *Figures 3*, *4* and *5* |
| Cell line (*Homo sapiens*) | HeLa cell line | ATCC | CCL-2 | *Figure 6* |
| Recombinant DNA reagent | pLIC_V2-*Dm-KLp61F motor- H6(1–369)* | This paper | | *Figure 1*, *2* |
| Recombinant DNA reagent | pLIC_V2-*Dm-KLP61F tail H6 (913–1016)* | This paper | | *Figure 1*, *2* |
| Recombinant DNA reagent | pLIC_V2-Dm KLP61F motor-tail fusion (residues 1–360, GSGSGS-linker, residues 913–1016) | This paper | | *Figure 1* |
| Recombinant DNA reagent | pET21a human Eg5 motor (residues 1–360) | This paper | Synthetic | *Figure 1* |

*Continued*

| Reagent type (species) or resource | Designation | Source or reference | Identifier | Additional Information |
|---|---|---|---|---|
| Recombinant DNA reagent | pET21a human Eg5 tail (residues 920–1056) | This paper | Synthetic | *Figure 1* |
| Recombinant DNA reagent | pET21a human Eg5 motor-tail fusion (residues 1–360 GSGSGS-linker residues 920–1056) | This paper | Synthetic | *Figure 1* |
| Recombinant DNA reagent | pFastbac-human FL-Eg5-GFP (residues 1–1056-msfGFP-StrepII) | This paper | | *Figure 3* |
| Recombinant DNA reagent | pFastbac-human FL-Eg5 (residues 1–1056-StrepII) | This paper | | *Figure 3* |
| Recombinant DNA reagent | pFastbac-human Eg5-Δ-tail-GFP (residues 1–920-msfGFP-StrepII) | This paper | | *Figure 3* |
| Recombinant DNA reagent | pcDNA3.1 FL-Eg5-mCh (residues 1–1056, mCherry) siRNA resistant (T2124C, C2130T, G2133T, and G2136A) | This paper | | *Figure 6* |
| Recombinant DNA reagent | pcDNA3.1 Eg5-Δtail-mCh (residues 1–920, mCherry) siRNA resistant (T2124C, C2130T, G2133T, and G2136A) | This paper | | *Figure 6* |
| Recombinant DNA reagent | GFP-Tubulin | Clonetech | Stock #61171 | *Figure 6* |
| Recombinant DNA reagent | pCMV-mCh | *Peris et al., 2009* | | |
| Peptide, recombinant protein | αβ-tubulin purified from porcine brains | This paper *Castoldi and Popov, 2003* | | *Figures 1*, *3* and *4* |
| Software, algorithm | ImageLab | Biorad | https://www.bio-rad.com/webroot/web/pdf/lsr/literature/10000076953.pdf | |
| Software, algorithm | FIJI (ImageJ) | *Schindelin et al., 2012* | https://fiji.sc | |
| Software, algorithm | Prism | GraphPad | https://www.graphpad.com/scientific-software/prism/ | |
| Software, algorithm | Motioncor2 | *Zheng et al., 2017* | https://emcore.ucsf.edu/ucsf-motioncor2 | |
| Software, algorithm | CTFFIND4 | *Rohou and Grigorieff, 2015* | https://grigoriefflab.umassmed.edu/ctf_estimation_ctffind_ctftilt | |
| Software, algorithm | EMAN2 | *Tang et al., 2007* | http://blake.bcm.edu/emanwiki/EMAN2 | |
| Software, algorithm | FREALIGN | *Grigorieff, 2007* | https://grigoriefflab.umassmed.edu/frealign | |
| Software, algorithm | B-factor | *Grigorieff, 2007* | https://grigoriefflab.umassmed.edu/bfactor | |
| Software, algorithm | UCSF-Chimera | *Pettersen et al., 2004* | https://www.cgl.ucsf.edu/chimera/ | |
| Software, algorithm | CCP4 suite | *Collaborative Computational Project, Number 4, 1994* | http://www.ccp4.ac.uk/html/dmmulti.html | |

*Continued on next page*

*Continued*

| Reagent type (species) or resource | Designation | Source or reference | Identifier | Additional Information |
|---|---|---|---|---|
| Software, algorithm | GCTF | *Zhang, 2016* | https://www.mrc-lmb.cam.ac.uk/kzhang/ | |
| Software, algorithm | Phyre protein homology model | *Kelley et al., 2015* | www.sbg.bio.ic.ac.uk/phyre2/html/page.cgi?id=index | |
| Software, algorithm | Relion 2.2 | *Emsley et al., 2010* | https://www2.mrc-lmb.cam.ac.uk/relion/index.php | |
| Software, algorithm | MolProbity | *Chen et al., 2010* | http://molprobity.biochem.duke.edu | |
| Software, algorithm | Coot | *Emsley et al., 2010* | http://www2.mrc-lmb.cam.ac.uk/personal/pemsley/coot/ | |

## Generation of constructs for various studies

For in vitro motility studies, we generated constructs for expression insect cells full length coding regions for Human Eg5 (KIF11)(FL-Eg5: residues 1–1056) and C-terminally truncated Eg5 (Eg5-Δtail; residues 1–920) were inserted in pFastbac1 vector in frame of a C-terminal StrepII tag or fused to monomeric superfolder-GFP(msf-GFP) and a C-terminal Strep-II tag.

For in vivo imaging studies, A pcDNA-3.1 plasmid containing Eg5 full-length fused to mCherry (Eg5 FL-mCh) inserted between the EcoR1 and Not1 sites was used as a template. Plasmid pAT4206, encoding Eg5-Δtail, which encoded residues 1–912, fused to mCherry (Eg5 ΔTail-mCh), was generated using the Phusion Site-Directed Mutagenesis Kit (Thermo Scientific). These clones contained silent mutations for siRNA-resistance: T2124C, C2130T, G2133T, and G2136A. The forward and reverse primer sequences used to generate Eg5 ΔTail-mCh were 5'-GGAGCGCCAATGGTGAGCAA-3' and 5'-AAAGCAATTAAGCTTAGTCAAACCAATTTT-3', respectively.

For Kinetic and structural studies, coding regions for isolated Dm KLP61F and Human Eg5 motor constructs (KLP61F motor residues 1–369; Eg5 motor residues 1–360) and the tail domains for these proteins (KLP61F tail residues 906–1016 and Eg5 residues 920–1056) and KLP61F motor-tail fusion (KLP61F residues 1–369 fused to 906–1016 and Eg5 residues 1–360 fused to residues 920–1056) were inserted in frame with histidine-tag in pET v2 vector (macrolab UC-berkeley). For affinity co-purification studies, a KLP61F tail region (residues 906–1016) was fused to a C-terminal StrepII tag.

## Protein expression and purification

For expression FL-Eg5-GFP, FL-Eg5 and Eg5-Δtail-GFP was carried out using baculoviral expression system. Briefly 200 to 500 mL of Spodoptera frugiperda (Sf9) cells, were infected with third passaged virus for each construct and expressed for 60–72 hr. Virus-Infected Sf9 Cells were centrifuged at 1500 rpm and then washed and incubated with lysis buffer (50 mM HEPES 300 mM KCl, 10 mM beta-meractoptoethanol, 1 mM MgCl2, 0.2 mM ATP) in the presence of 0.5% Triton X100. Cells were lysed by extrusion using a dounce homogenizer and then were centrifuged at 40,000 rpm using Ti45 rotor in ultracentrifuge (Beckman). The lysate was passaged on a Streptactin XT resin (IBA lifesciences) equilibrated with lysis buffer, washed extensively and then eluted with lysis buffer supplemented with 100 mM D-Biotin. Purified Eg5 proteins were concentrated and loaded onto Superose-6 (10/300) column (GE Health Care) equilibrated onto AKTA system and eluted in 0.5 mL fractions (*Figure 3—figure supplement 1A–B*). Fractions were evaluated using SDS-PAGE for protein quality and Eg5 protein containing fractions were either used immediately for motility experiments or were snap frozen with 15% glycerol in lysis buffer using liquid nitrogen. We did not observe any difference in the FL-Eg5-GFP, FL-Eg5 and Eg5-Δtail-GFP activities in the frozen or freshly prepared settings (*Figure 3—figure supplement 1A–B*).

For kinetic and structural cryo-EM studies, Dm-KLP61F and human Eg5 motor, tail and motor-tail constructs were expressed using T7 expression in SoluBL21 cells. Briefly, cells were grown at 37°C and induced for expression using 0.5 mM Isopropyl thio-beta-glucoside overnight at 19°C. Cells were centrifuged and lysed in Lysis buffer using a microfluidizer. Proteins were purified using IDA-nickel affinity chromatography (Macherey Nagel), washed extensively and then eluted with lysis buffer supplemented with 200 mM Imidazole. The purified fractions were loaded onto a Superdex

200 (16/6) size exclusion column equilibrated with lysis buffer at 1 mL fractions. SDS-PAGE was used to evaluate fractions and concentrated pure proteins were incubated with 15% glycerol before freezing in liquid nitrogen.

## MT stimulated ATP hydrolysis, tail affinity motor co-purification and MT-co-sedimentation assays

MT-stimulated ATPase activity for KLP61F and Eg5 motor, tail, motor-tail fusion and Eg5-motor and tail constructs were carried out by measuring the free phosphate production rate in 50 mM Potassium acetate (K-acetate) or 20 mM KCl, 25 mM HEPES, 5 mM magnesium acetate, 1 mM EGTA, pH 7.5 buffer in the presence of a minimum of a 5-fold molar excess range of MT concentrations, using a commercially-available kit (EnzChek, Molecular Probes) at 20° C. The Eg5 motor, tail and motor-tail fusion were additionally studied at 20 mM KCl due to the 10-fold higher Eg5 motor MT stimulated ATP hydrolysis $K_m$ in 50 mM K-Acetate. These comparisons suggest that the Eg5 tail regulation of motor ATPase is slightly weaker at 50 mM K-Acetate conditions (*Figure 1—figure supplement 1L–M*).

For affinity co-purification for the motor and tail, 5–10 µmoles of KLP61F tail-StrepII tagged protein was incubated with 5–10 µmoles of the KLP61F motor domain in incubation buffer (25 mM HEPES, 25–75 mM KCl 5 mM MgCl$_2$ and 1 mM EGTA, 2% glycerol). The mixture (Load) was then incubated with 200 µl of StrepXT resin (IBA-biotech). The flow-through fraction from the resin was collected (FL). The resin was washed with two column volumes of incubation buffer. The bound fraction was then eluted (Elute) with incubation buffer + 50 mM biotin. The mixture was analyzed by SDS PAGE. Quantitative densitometry was carried out to obtain the motor/tail molar ratio using Biorad Image Lab software (Biorad). Motor and tail band intensities were measured, and the intensity data was converted to µmol values and then molar ratios were calculated (*Figure 1E*, *Figure 1—figure supplement 1A*).

To measure MT co-sedimentation activity for KLP61F and Eg5 motor and tail constructs, we prepared 5 mg/ml MTs in 5% DMSO by polymerization at 37 °C and then stabilized with Paclitaxel (sigma), MT co-sedimentation assays were carried out by mixing 0.01, 0.025, 0.05, 0.1, 0.25 up to 0.05 µmol of KLP61F or Eg5 motor with or without their respective tail domains, in 25 mM HEPES pH 7.5, 5 mM MgCl2, 1% glycerol, 25 mM or 75 mM KCl in the presence of 2 mM AMPPNP, 2 mM ADP, 2 mM ADP.AlF4 or 10U Apyrase to mimic the nucleotide-free state. The five to six different concentration mixtures of motors, tail indicated above were mixed with MTs in these conditions were incubated at 25°C for 20 min and then centrifuged at 18 k for 25 min at 25 °C. A control condition is usually included in which co-sedimentation is carried out without MTs (-MTs). The supernatant fractions were then removed and mixed with SDS sample buffer. The pellets were resuspended with SDS sample buffer. Equal amounts of each supernatant and pellet fractions for 0.01, 0.025, 0.05, 0.1, 0.25 µmol and -MT conditions were analyzed by SDS PAGE (*Figure 1F–H*; *Figure 1—figure supplement 1*),. Quantitative densitometry was carried out using Biorad Image Lab software (Biorad) to determine the saturated molar ratios of the motor and tail to polymerized tubulin in MTs. The intensities for each band was measured in comparison to a local background control. The intensity data were converted to µmoles and then molar ratios were calculated for each lane average ratios were calculated for all saturated lanes.

## Sample preparation of motor and tail decorated MTs for Cryo-EM

MTs were prepared by polymerizing 5 mg/ml tubulin (Cytoskeleton, Denver, CO) in BRB80 buffer (80 mM PIPES, pH 6.8, 1 mM EGTA, 4 mM MgCl2, 2 mM GTP, 9% dimethyl sulfoxide) for 30 min at 36° C. Paclitaxel was added at 250 µM before further incubation of 30 min at 36°C. The polymerized MTs were then incubated at room temperature for several hours or overnight before use. For grid preparations, KLP61F motor and KLP61F tail proteins were mixed in 0.5% binding buffer (25 mM HEPES, 35 mM potassium acetate plus 2 mM ADP to a final concentration of 0.1 mg/mL and 0.04 mg/mL respectively). For grid preparation of motor alone KLP61F motor (10 mg/mL) was diluted in binding buffer (50 mM HEPES, 70 mM potassium acetate plus 2 mM AMPNP). All samples were prepared on 1.2/1.3 400-mesh grids (Electron Microscopy Services). Grids were glow-discharged before sample application. The cryo-samples were prepared using a manual plunger, which was placed in a homemade humidity chamber that varied between 80% and 90% relative humidity. A 4 µl amount of

the MTs at ~0.5 µM in 80 mM PIPES, pH 6.8, 4 mM MgCl2, and 1 mM EGTA supplemented with 20 µM Paclitaxel was allowed to absorb for 2 min, and then 4 µl of the KLP61F motor and tail domains were added to the grid. After a short incubation of 2 min, 0.5 µL of Apyrase (25 units) was added and after a short incubation of 3 min the sample was blotted (from the back side of the grid) and plunged into liquid ethane. This procedure was repeated for the motor alone preparations without the addition of the apyrase step.

## Cryo-EM image analysis, structure determination and model building

Images of frozen-hydrated KLP61F motor decorated MTs in the AMPPNP state or KLP61F motor and tail decorated MTs in the nucleotide-free state (see *Table 2*) were collected on a Titan Krios (FEI, Hillsboro, OR) operating at 300 keV equipped with a K2 Summit direct electron detector (Gatan, Pleasanton, CA). The data were acquired using the Leginon automated data acquisition (*Suloway et al., 2005*). Image processing was performed within the Appion processing environment (*Lander et al., 2009*). Movies were collected at a nominal magnification of 22500 × with a physical pixel size of 1.31 Å/pixel. Movies were acquired using a dose rate of ~7.96 and 8.3 electrons/pixel/second over 8.25 s yielding a cumulative dose of ~38 and 40 electrons/Å2 (respectively). The Motion-Cor frame alignment program (*Hirschi et al., 2017*; *Li et al., 2013*) was used to motion-correct. Aligned images were used for CTF determination using CTFFIND4 (*Rohou and Grigorieff, 2015*) and only micrographs yielding CC estimates better than 0.5 at 4 Å resolution were kept. MT segments were manually selected, and overlapping segments were extracted with a spacing of 80 Å along the filament. Binned boxed segments (2.62 Å/pixel, 192 pixel box size) were then subjected to reference-free 2D classification using multivariate statistical analysis (MSA) and multi-reference alignment (MRA) (*Hirschi et al., 2017*). Particles in classes that did not clearly show an 80 Å layer line were excluded from further processing.

For cryo-EM reconstruction, undecorated 13,14- and 15-protofilament MT densities (*Sui and Downing, 2010*) were used as initial models for all preliminary reconstructions. We used the IHRSR procedure (*Egelman, 2007*) for multi-model projection matching of MT specimens with various numbers of protofilaments (*Alushin et al., 2014*), using libraries from the EMAN2 image processing package (*Tang et al., 2007*). After each round of projection matching, an asymmetric back-projection is generated of aligned segments, and the helical parameters (rise and twist) describing the monomeric tubulin lattice are calculated. These helical parameters are used to generate and average 13, 14 and 15 symmetry-related copies of the asymmetric reconstruction, and the resulting models were used for projection matching during the next round of refinement. The number of particles falling into the different helical families varied. Helical families that had enough segments were further refined. Final refinement of MT segment alignment parameters was performed in FREALIGN (*Grigorieff, 2007* without further refinement of helical parameters. FSC curves were used to estimate the resolution of each reconstruction, using a cutoff of 0.143. To better display the high-resolution features, we applied a B-factor of 200 Å, using the program bfactor (http://grigorieﬄab.janelia.org). The final statistics for all data sets were described in *Table 2*.

In order to enhance the total mass of the mobile tail density and improve the resolution of the MT decorated with KLP61F motor and tail in the nucleotide-free state, an additional round of 'MT-patch refinement' processing was performed enhance conformational homogeneity. The same motion-corrected micrographs and boxes were used, but defocus parameters were re-estimated using GCTF (*Zhang, 2016*). MTs were then sorted into 13, 14, and 15 protofilament MTs using reference alignment as previously described (*Shang et al., 2014*). Of the 29,274 starting particles, roughly two thirds (19,128) MT particles corresponding to the 14 protofilament symmetry were selected for further processing. MTs were then refined using RELION helical processing (*He and Scheres, 2017*). Initially, the asymmetric unit was defined as one full 82 Å repeat (consisting of 13 tubulin dimers), using an initial estimate of zero for the helical twist. Local symmetry searches were performed during the refinement to optimize these parameters. Following refinement, the particle coordinates were smoothed as previously described (*Huehn et al., 2018*). After MT refinement, an additional protofilament refinement step was performed in an attempt to increase resolution of the final volume by correcting for distortions in the MT lattice. To do this, a wedge mask is applied to the final MT volume, resulting in a MT missing a single protofilament. This volume was then rotated and subtracted thirteen times from each experimental image to generate a stack of protofilament particles, with one particle for every tubulin dimer in the imaged filament. In this case, 267,792

protofilament particles were obtained from the original stack of 19,128 MT particles. Protofilament particles alignment parameters were initialized using Euler angles derived from the MT refinement step and subjected to further, local refinement using RELION. The final resolution was computed using the RELION post-processing module with a soft-edged mask. A more detailed description of the protofilament refinement method is currently being prepared for publication (Debs et al. manuscript in preparation). In order to more accurately estimate the resolution of each region of the reconstructed density, a local resolution calculation was performed using the 'blocres' function in the Bsoft processing package (*Heymann and Belnap, 2007*). This analysis revealed that the majority of the tubulin density is in the range of 3.5–4.5 Å, while the kinesin portion ranges from 5 to 6 Å resolution and tail density is around 8 Å resolution (*Figure 2—figure supplement 1A-C*). Model building was performed using the programs Coot and UCSF chimera using the kinesin-5 structural model (*Turner et al., 2001*). All the ligands were included in the model except the taxol molecule. The model was adjusted using the secondary structure elements in the density maps and analyzed for clash score using Coot and Molprobity (*Table 3*). The maps were compared using the programs UCSF chimera and Coot to determine the transitions of various elements.

## Reconstitution of human Eg5 motility along single MTs

Kinesin-5 MT stimulated motility was reconstituted as follows: Flow chambers were assembled from N 1.5 glass coverslips (0.16 mm thick; Ted Pella) that were cleaned with the Piranha protocol and functionalized with 2 mg/mL PEG-2000-silane containing 2 µg/mL biotin-PEG-3400-silane (Laysan Bio) suspended in 80% at pH 1 (*Henty-Ridilla et al., 2016*). After the flow chamber was assembled, 0.1 mg/mL NeutrAvidin (Thermofisher) was used to functionalize surfaces. Biotin and Alexa-Fluor-633-labeled porcine tubulin were generated in the laboratory as described (*Al-Bassam, 2014*) and were polymerized using the non-hydrolysable GTP analog guanosine-5'-[(α,β)-methyleno] triphosphate (termed GMPCPP; Jena Biosciences) or using the MT stabilizing drug, Paclitaxel (sigma). These MTs (100–200 µg/mL in BRB-80: 80 mM PIPES, 1 mM $MgCl_2$ and 1 mM ETGA; pH 6.8, 1% glycerol, 0.5% pluronic-F127, 0.3 mg/ml casein, 3 mM BME, 4 mM ATP-MgCl2) were flowed into chambers and attached to glass via biotin-neutravidin linkage. Flow chambers were then extensively washed with imaging buffer (25 mM HEPES, 25–100 mM KCl, pH 7.5, 10 mM beta-mercatopethanol; 1% glycerol, 0.5% Pluronic-F127, 0.3 mg/ml casein, 3 mM BME, 4 mM ATP-MgCl2). Kinesin-5 MT-stimulated motility was reconstituted at 25°C by injecting 1–20 nM FL-Eg5-GFP combined with a photobleach-correction mixture into flow chambers (*Telley et al., 2011*). Movies were captured in TIRF mode using a Nikon Eclipse Ti microscope using 1.5 Na objective and an Andor IXon3 EM-CCD operating with three (488 nm, 560 nm and 640 nm) emission filters using alternating filter wheel in 2 s increments operated using elements software (Nikon).

## Reconstitution of human Eg5 MT sliding motility

To study MT sliding activities in vitro, flow chambers were prepared as described above and either Paclitaxel or GMPCPP stabilized AlexaF-633 and biotin labeled MTs were anchored along their surface via Biotin-Neutravidin linkage. A mixture of 1–20 nM FL-Eg5-GFP or 20–200 nM Eg5-Δtail-GFP were mixed with 100–200 µg/ml AlexaF-560 labeled MTs and injected into these flow chambers, after being equilibrated with imaging buffer. Imaging was initiated as described above almost immediately and areas of MT sliding events were identified through search. At 3–20 nM FL-Eg5-GFP robust free MT crosslinking (yellow) was observed, followed by zippering along anchored MT (red) and then MT sliding was consistently observed (*Figure 4B*, *Videos 2–3*). In contrast, 3–20 nM Eg5-Δtail-GFP MT crosslinking was often observed but free-MTs (yellow) did not zipper along the anchored MT (red) and remain in scissoring motion for extensive periods (*Figure 4B*).

For MT sliding and kinesin-5 spiking studies, MT sliding experiments were performed as described above with the exception of using 20 nM FL-Eg5. Eg5-GFP spiking was carried out by the addition of either 1 nM FL-Eg5-GFP or 1 nM Eg5-Δtail-GFP with 100–200 µg/ml AlexaF-560-MTs in imaging buffer conditions. TIRF Imaging was carried out as described above.

## Image analysis of motor motility and MT sliding motility

Image movie stacks were preprocessed with photobleach correction and image stabilization plugins using the program FIJI (*Schindelin et al., 2012*). For motility along individual MTs, individual FL-Eg5-

GFP or Eg5-Δtail-GFP motor motility events were identified along anchored MTs based on kymographs in generated for multiple channels. The FIJI plugin, trackmate, (*Schindelin et al., 2012*) was used to measure particle motility rates and identify their run lengths. Large collections of motile events for FL-Eg5-GFP or Eg5-Δtail-GFP conditions were collected for 25, 50, and 100 mM KCl conditions (*Table 4*). Average MT parameters were determined by frequency binning the motility events in a range conditions and then fitting these events using Gaussian distributions using the program Prism (*Table 4*). In general, all parameters fit single Gaussian distributions. Run lengths were fitted using exponential decay to identify the half-length for each motor condition. T-tests were performed to determine significance of the differences observed.

For motor fluorescence intensity quantifications, kymographs were manually analyzed using the line tool in FIJI, a line was placed over the initial signal of an individual Eg5 molecule and an intensity profile was generated and recorded in Microsoft Excel. The line was extended to include an area of the kymograph where a fluorescent signal was absent in order to measure the local surrounding background signal. This background measurement was subtracted from the initial fluorescence intensity of the molecules signal in Microsoft Excel. Only molecules that were observed to have landed on the MT during the observation period, and that were motile were used for quantification. The intensity data were frequency binned and Gaussian fit using Prism.

For MT sliding and MT sliding spiking assays, image analysis was carried out as described above with the exception of visualizing the free-MT sliding motility with respect to the anchored MT using 560 nm emission channel. The motility patterns FL-Eg5-GFP or Eg5-Δtail-GFP motor particles were studied with respect to sliding zone (along both the free and anchored MT) using the FIJI plugin, Trackmate, to determine motor velocities and their changes in motility direction inside or outside the sliding zone.

## Measuring pushing forces by optical trapping human Eg5 MT sliding events

To study Eg5 MT sliding forces in optical trapping, flow chambers were prepared as described above and previously (*Shimamoto et al., 2015*). Paclitaxel stabilized Hilyte-649 and biotin labeled MTs were anchored along their surface via Biotin-Neutravidin attachment. Polystyrene beads were coated with kinesin-1 nucleotide-free mutant and linked to Rhodamine labeled MTs (bead attached free-MTs). These bead attached free-MTs were then mixed with 1–20 nM FL-Eg5-GFP or 1–500 nM Eg5-Δtail-GFP and injected into these flow chambers, after being equilibrated with imaging buffer. The beads attached free MTs were observed to interact with the anchored MTs and locked into the optical trap to measure the forces. Generally, 3–10 nM FL-Eg5-GFP was sufficient to observe MT sliding events, while 3–10 nM Eg5-Δtail-GFP rarely produced sliding events, and mostly crosslinked without zippering into sliding zones forming scissoring events. At 200–500 nM Eg5-Δtail-GFP, we observed sufficient MT sliding events. For each event, the length of the sliding zone, the total Eg5-GFP intensity, and plateau pushing forces developed were measured and used for scaled comparisons (*Figure 5B*). The Eg5-GFP intensity scaled linearly with the MT overlap MT sliding zone length without significant difference between FL-Eg5-GFP or Eg5-Δtail-GFP conditions, despite the difference in the concentration using in the assay (*Figure 5D*). The plateau forces generated by FL-Eg5-GFP scaled linearly with the size of the overlap MT sliding zone length, with seven-fold difference in the slope of the same comparison for Eg5-Δtail-GFP data.

## Transfection of hela cells and in vivo imaging

HeLa cells were cultured in Minimal Essential Media-α (Gibco) supplemented with 10% fetal bovine serum (Gibco). HeLa cells were authenticated by STR genotyping by the Vermont Integrative Genomics Resource. Cells were maintained at 37°C with 5% $CO_2$. Transient transfections of plasmid DNA were performed via electroporation using a Nucleofector 4D system, pulse code CN114, and Cell Line SE reagents (Lonza). Cells were plated onto 12 mm glass coverslips (Electron Microscopy Sciences) for fixed cell immunofluorescence, 4-chamber 35 mm glass-bottom dishes (Greiner Bio-One) for live cell imaging, or 60 mm polystyrene tissue culture dishes for lysate collection.

To assess the protein levels in vivo, cells were arrested in 100 μM monastrol (Selleckchem) overnight and lysed in PHEM buffer (60 mM PIPES, 25 mM HEPES, 10 mM EGTA, 4 mM MgSO₄) with Halt Protease and Phosphatase Inhibitor cocktail (Thermo-Fisher) on ice. Lysates were extracted on

ice for 10 min and centrifuged at 21,130 x g for 10 min. An equal volume of 4X Laemmli buffer (Bio-Rad) was added to the supernatant and samples were heated to 95°C for 10 min. Lysates were separated by electrophoresis on 4–15% Tris-glycine polyacrylamide gels (Bio-Rad) and transferred to polyvinylidene fluoride membranes (Bio-Rad). Membranes were blocked in Odyssey blocking reagent (LI-COR) diluted 1:1 in tris-buffered saline for 1 hr, incubated with rabbit anti-mCh (diluted 1:1000, AbCam) and mouse anti-GAPDH (diluted 1:10,000, Thermo-Fisher) primary antibodies overnight, and incubated with IRDye 800- and IRDye 680-tagged fluorescent secondary antibodies (LI-COR) for 1 hr. Blot fluorescence was imaged using an Odyssey CLx system (LI-COR) and analyzed using Image Studio Lite (LI-COR).

Fixed and live cell imaging was performed using a Nikon Ti-E inverted microscope controlled by NIS Elements software (Nikon Instruments) with a Plan APO 60X/1.42 NA oil immersion objective or APO 100X/1.49 NA oil immersion objective (Nikon Instruments), Spectra-X light engine (Lumencore), and Clara CCD camera (Andor). Image processing was performed using NIS Elements (Nikon Instruments) and ImageJ (NIH). Data analysis and statistical comparisons were performed using Excel (Microsoft) and Prism (GraphPad Software).

For assessment of mCh and Eg5-mCh expression levels and localization at metaphase in fixed HeLa cells, cells were treated with 20 μM MG132 (Selleckchem) 2 hr prior to fixation. Cells were fixed for 10 min in 1% paraformaldehyde (Electron Microscopy Sciences) in ice-cold methanol (Thermo-Fisher). Cells were blocked using 20% goat serum in antibody-diluting buffer (AbDil, 1X tris-buffered saline with 2% bovine serum albumin, 0.1% Triton-X 100, and 0.1% sodium azide) for 1 hr, incubated with mouse anti-α-tubulin primary antibodies (DM1a, Sigma-Aldrich, diluted 1:750 in AbDil) for 1 hr, and incubated in fluorescent secondary antibodies conjugated to Alexa Fluor 488 or 647 (Life Technologies, diluted 1:500 in AbDil) for 1 hr. Cells were mounted in ProLong Gold with DAPI (Thermo-Fisher). Expression levels of mCh-tagged proteins were compared by drawing elliptical regions of interest (ROIs) around mitotic cells using α-tubulin staining, measuring mCh fluorescence intensity within the cellular ROIs, and subtracting averaged intensity from two background ROIs containing no visible cells. Localization was assessed by defining an ROI as the spindle based on α-tubulin staining and an ROI as cytoplasm by subtracting this spindle ROI from an ellipse that encompassed the cell. Intensity of mCh signal was measured in both the spindle and cytoplasm ROIs, and a ratio of spindle to cytoplasm intensity calculated.

For live cell imaging, growth media was exchanged for $CO_2$-Independent Media (Gibco) supplemented with 10% fetal bovine serum (Gibco) and penicillin/streptomycin (Gibco). For assessment of localization after treatment with BRD-9876 (Tocris Bioscience), HeLa cells were treated with 20 μM MG132 (Selleckchem) for 2 hr prior to imaging. Cells were imaged prior to drug addition, 1 min after addition of 5 μM BRD-9876, and subsequently once every 5 min. For assessment of localization in anaphase, cells in metaphase were identified and imaged at 2 min intervals through anaphase. For both live cell assays, localization of proteins to the spindle was quantified as described for fixed cell imaging, with the spindle ROI defined by GFP-tubulin signal.

## Acknowledgements

We thank Dr Jonathan Scholey (Molecular Cellular Biology) for inspiring JAB with excitement about studying kinesin-5 mechanism. JAB is supported by funding support from the NSF- 1615991 and NIH-GM110283. RM. is supported by funding from NIH-GM052468. RJM is supported by funding from the NIH- GM124889. SR is supported by funding from NIH-GM130556. LG is supported in part by funding from the Israel Science Foundation (ISF) (ISF 386/18), and US NSF-Israel Binational science foundation (BSF-2015851). JS is supported by funds from the NIH-GM121491 and NIH-GM130556.

## Additional information

### Funding

| Funder | Grant reference number | Author |
|---|---|---|
| National Science Foundation | 1615991 | Jawdat Al-Bassam |

| National Institutes of Health | GM110283 | Jawdat Al-Bassam |
| National Institutes of Health | GM121491 | Jason Stumpff |
| National Institutes of Health | GM130556 | Steven S Rosenfeld Jason Stumpff |
| Israel Science Foundation | ISF 386/18 | Larisa Gheber |
| United States-Israel Binational Science Foundation | BSF-2015851 | Larisa Gheber |
| National Institutes of Health | GM052468 | Ron Milligan |
| National Institutes of Health | GM124889 | Richard J McKenney |

The funders had no role in study design, data collection and interpretation, or the decision to submit the work for publication.

### Author contributions

Tatyana Bodrug, Steven S Rosenfeld, Data curation, Formal analysis, Funding acquisition, Validation, Investigation, Visualization, Methodology; Elizabeth M Wilson-Kubalek, Data curation, Software, Formal analysis, Supervision, Funding acquisition, Validation, Investigation, Visualization; Stanley Nithianantham, Data curation, Formal analysis, Supervision, Funding acquisition, Validation, Investigation, Visualization; Alex F Thompson, Data curation, Formal analysis, Funding acquisition, Investigation, Visualization; April Alfieri, Ignas Gaska, Jennifer Major, Data curation, Formal analysis, Funding acquisition, Investigation; Garrett Debs, Formal analysis, Investigation, Methodology; Sayaka Inagaki, Data curation, Investigation; Pedro Gutierrez, Charles Vaughn Sindelar, Formal analysis, Investigation; Larisa Gheber, Scott T Forth, Data curation, Formal analysis, Investigation, Visualization, Methodology; Richard J McKenney, Data curation, Formal analysis, Investigation; Ronald Milligan, Data curation, Formal analysis, Funding acquisition; Jason Stumpff, Data curation, Formal analysis, Validation, Investigation, Methodology; Jawdat Al-Bassam, Conceptualization, Resources, Data curation, Software, Formal analysis, Supervision, Funding acquisition, Validation, Investigation, Visualization, Methodology, Writing - original draft, Project administration, Writing - review and editing

### Author ORCIDs

Tatyana Bodrug (iD) https://orcid.org/0000-0001-9017-962X
Stanley Nithianantham (iD) https://orcid.org/0000-0001-6238-647X
Larisa Gheber (iD) http://orcid.org/0000-0003-3759-4001
Charles Vaughn Sindelar (iD) http://orcid.org/0000-0002-6646-7776
Jawdat Al-Bassam (iD) https://orcid.org/0000-0001-6625-2102

### Decision letter and Author response

Decision letter https://doi.org/10.7554/eLife.51131.sa1
Author response https://doi.org/10.7554/eLife.51131.sa2

## Additional files

### Supplementary files

• Transparent reporting form

### Data availability

The atomic coordinate file for Dm-KLP61F motor AMPPNP MT(alpha-beta-tubulin) asymmetric unit model is available at Protein Data Bank (PDB-ID: 6VPO) The Dm-KLP61F motor nucleotide-free MT (alpha-beta-tubulin) asymmetric unit model is available at Protein Data Bank (PDB-ID: 6VPP). The refined Dm-KLP61F motor AMPPNP MT cryo-EM map is available at the Electron microscopy Data bank (EMDB ID:EMD-21314) and the Dm-KLP61F motor-tail nucleotide-free (focused classification map) MT cryo-EM map is available at Electron microscopy Data bank (EMDB-iD:EMD-2135).

The following datasets were generated:

| Author(s) | Year | Dataset title | Dataset URL | Database and Identifier |
|---|---|---|---|---|
| Wilson-Kubalek E, Nithianantham S, Al-Bassam J | 2020 | Atomic model for Klp61F motor domain AMPPNP in complex with alpha-beta tubulin microtubule asymmetric unit | https://www.rcsb.org/structure/6VPO | RCSB Protein Data Bank, 6VPO |
| Wilson-Kubalek E, Nithianantham S, Al-Bassam J | 2020 | Cryo-EM map for KLP61F motor domain AMPPNP in complex with alpha-beta tubulin microtubule asymmetric unit | https://www.ebi.ac.uk/pdbe/entry/emdb/EMD-21314 | Electron Microscopy Data Bank, EMD-21314 |
| Wilson-Kubalek E, Nithianantham S, Al-Bassam J | 2020 | Atomic model for Klp61F motor domain in the nucleotide free state in complex with alpha-beta-tubulin microtubule asymmetric unit | https://www.rcsb.org/structure/6VPP | RCSB Protein Data Bank, 6VPP |
| Wilson-Kubalek E, Nithianantham S, Al-Bassam J | 2020 | Cryo-EM map (locally classified) for KLP61F motor and tail domain complex in the nucleotide free state with alpha-beta tubulin microtubule asymmetric unit | https://www.ebi.ac.uk/pdbe/entry/emdb/EMD-21315 | Electron Microscopy Data Bank, EMD-21315 |

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
