## [Decision Letter]

**Acceptance summary:**

Bodrug et al. use a predominantly in vitro approach to investigate the role of the Kinesin-5 tail in regulating the motor's mechanochemical cycle and ability to slide anti-parallel microtubules. Using microtubule co-sedimentation and nucleotide activity assays, the authors find that the tail domain downregulates Kinesin-5 ATP hydrolysis. This finding is further corroborated using cryo-EM, where it is observed that the tail engages and opens up a nucleotide-free Kinesin-5 active site on microtubules. This conformation is predicted to reduce the motor's ability to bind incoming ATP. In line with this observation, Bodrug et al. show that tail engagement promotes slow motor motility and increases clustering of Kinesin-5 on microtubules, which in turn generates the force necessary for anti-parallel microtubule sliding. The combined biochemical, structural and cellular strategies adopted provide credence for the authors' revised model of Kinesin-5 regulation.

**Decision letter after peer review:**

Thank you for submitting your article "The kinesin-5 tail domain directly modulates the mechanochemical cycle of the motor for antiparallel microtubule sliding" for consideration by *eLife*. Your article has been reviewed by three peer reviewers, one of whom is a member of our Board of Reviewing Editors, and the evaluation has been overseen by Suzanne Pfeffer as the Senior Editor. The following individual involved in review of your submission has agreed to reveal their identity: Ryo Nitta (Reviewer #2).

The reviewers have discussed the reviews with one another and the Reviewing Editor has drafted this decision to help you prepare a revised submission.

Summary:

The authors investigate the fascinating role of kinesin-5 tail for antiparallel microtubule sliding by biochemical, structural, and biophysical approaches. They proposed that the tail domain affects motor functions including the speed of the motility, microtubule-binding affinity, or the force production, coupled with the nucleotide state of the motor. Cryo-EM structure of the MT-kinesin-5 clearly visualizes the additional density (maybe, tail domain) on the motor domain (not on the microtubule). MT sliding assays and optical trapping illustrate the importance of the tail domain for the antiparallel microtubule sliding to accomplish the mitotic function of kinesin-5.

Overall, experimental data provide the intriguing regulation mechanism of kinesin-5 motility by its tail. The reviewers were all supportive of asking for revisions. The feeling was that the authors need to improve the clarity and accuracy of this manuscript during revision.

Essential revisions:

Reviewer 1:

1) All figure captions, where applicable, need to state number of experimental repeats, type of statistics performed, how many molecules were included, units of average speed above histograms, axis units on inset graphs and brief experimental details. For example, Figure 1B-D, H, Figure 3F-G, Figure 4C-D, H, I, Figure 5D-F, Figure 6E.

2) No model refinement parameters have been presented. Tables of refinement statistics is required for each model generated from the cryo-EM experiments.

3) FSC curve on Figure 2—figure supplement 1C(ii) shows a sharp drop-off (followed by a rise) at the low spatial frequency (~0.10 1/Å), usually indicative of an applied mask that is too tight. Can the authors try a looser mask?

Reviewer 2:

1) Exploring direct binding between the motor and tail domains:

Motor-tail interaction is the key feature of this manuscript. The authors performed microtubule co-sedimentation assays as well as cryo-EM analysis to propose that the tail domain of KLP61F is not bound to the microtubule but is bound directly to the motor domain through an α-helix to regulate the catalytic cycle of the motor. However, biochemical assays did not show the direct binding between motors and tails without microtubules. Cryo-EM studies illustrate the additional density at the minus-end side of the motor, albeit it is much smaller than the expected size of the tail at ~8 Å resolution. Thus, it is not strong enough to conclude whether the additional density is actually the tail or not. The reviewers agree that a direct biochemical interaction between motor and tail should be tested without microtubules (e.g. by: GST-pull down assay, Gel-filtration (or SEC-MALS), Calorimetry, Biacore, etc).

2) Validating the motor-tail interface supposed by cryo-EM structure:

Related to comment 1, a mutagenesis experiment will strengthen the structural model proposed. Please introduce a point mutation in α0-helix to investigate the contribution of this helix to the binding to the tail domain.

3) Checking the functional identity between KLP61F and Eg5:

In the manuscript, the first half (biochemical and structural studies) was mostly done with *Drosophila* KLP61F, whereas the other half (biophysical studies) was done with human Eg5. The authors connect these stories together with the assumption that the role of the KLP61F and Eg5 are the same. Based on the sequence identity between KLP61F and Eg5, overall identity is 33%, and that of the motor domain is more than 50%, however, those of the tail and α 0 are around 20% , which is not high enough to easily conclude they play the same function. To overcome the gap, the authors need to experimentally prove functional identity between KLP61F and Eg5, including the motor-tail binding and the regulation of ATPase activity by tail under the same conditions (nucleotide analog, salt concentration, counter ion, etc.).

4) Decreasing the ATPase rate and the velocity with/without tail:

The authors claim that the tail domain of kinesin-5 down-regulates the microtubule-activated ATPase activity of the motor, resulting in the slow motility of kinesin-5. In Table 1 and Figure 1, the ATPase rate of Eg5 was decreased to half by the addition of the tail. On the other hand, in Table 3 and Figure 3, the velocity of Eg5 was decreased to one-quarter or one-fifth by the addition of the tail. How can authors explain this discrepancy? Is there some additional reason of slow motility?

Reviewer 3:

1) The authors need to significantly improve the clarity and accuracy of this manuscript in the revision, as the writing of the current version does not measure up to the quality of the work and the current manuscript contains surprisingly too many errors that have made the manuscript somewhat difficult to read. Some essential errors are listed below:

1A) Panel E is missing in Figure 1; as a result, references to many panels of Figure 1 in the text are incorrect; and the text contains no mentioning of Figure 1J.

1B) In the first paragraph of the subsection “The kinesin-5 tail domain downregulates MT activated ATP hydrolysis by binding the motor domain in the ADP or nucleotide-free states”, it should be "1-369" instead of "1-356", based on the labeling in Figure 1A.

1C) "Figure 3A, lower panel" should be "Figure 3C".

1D) “Figure 3—figure supplement 1C should be “Figure 3—figure supplement 1D-E”.

1E) “Figure 3C” should be “Figure 3—figure supplement 1C”.

2) Based on the GFP fluorescence intensity distribution of motile Eg5-Δtail-GFP and FL-Eg5-GFP, the authors state in the text (e.g. subsections “The tail domain down-regulates kinesin-5 motility velocity along single MTs”, third paragraph and “A modified model for kinesin-5 tail regulated hand-over-hand motility and its essential role in antiparallel MT sliding activity”, last paragraph) and in Figure 7A that in higher ionic strength buffers the tail of kinesin-5 may be involved in inter-molecular interaction to cluster multiple kinesin-5 homotetramers into higher-order oligomeric complexes. Given that in TIRF microscopy experiments, GFP fluorescence intensity of these motility Eg5 particles highly depends on experimental settings, e.g., laser intensity and location in the field of view, the authors need to use one other independent assay such as sucrose gradient centrifugation to verify whether FL-Eg5-GFP indeed exists mainly as high-order oligomeric complexes in high ionic strength buffers.

3) While the authors have demonstrated that the tail domain slows kinesin-5 velocity both along single MTs and within the overlap zone formed between two antiparallel MTs, it is unclear whether this tail-to-motor interaction indeed forms part of an active regulatory pathway in vivo. As such, in the Results, the authors need change "downregulate" and "down-regulate" (or their likes) to reduce/decrease to focus on describing rather than interpreting their experimental observations.

4) In the second paragraph of the Introduction, the authors state that yeast kinesin-5s reverse direction toward the plus-ends upon clustering into multi-motor complexes along single MTs. Which study (or studies) are the authors referring to?

5) It is inaccurate to state that in motility buffers contains 25-100mM KCl, "[t]his processive motility is highly homogeneous", given that the velocity histograms for FL-Eg5-GFP and Eg5-∆tail-GFP at 100 mM KCl both contain two different modes.

6) In the subsection “The tail domain is critical for slowing kinesin-5 motility within MT sliding zones”, the description of FL-Eg5-GFP motility is inaccurate, and it is not obvious from Figure 4F that FL-Eg5-GFP exhibits slower plus-end directed motility within the MT overlap zone than on the surface-anchored MT outside the overlap zone; judging by the inset velocity histograms in Figure 4H, it is unclear whether the difference between the two velocities is statistically significant.

7) In the subsection “The tail domain is critical for slowing kinesin-5 motility within MT sliding zones”, it is not clear to me how these data would necessarily suggest the kinesin-5 tail regulates MT-sliding rate by modulating a unique motor association with MTs and coupling between two MT-bound ends of kinesin-5. How about the presence of the tail domain prolonging the retention of kinesin-5 within the MT-binding overlap zone by being an extra MT-binding site?

8) In the Discussion, the authors state in multiple places that the tail domain regulates processive hand-over-hand stepping during kinesin-5 motility. Given that it remains to be determined whether homotetrameric kinesin-5s use hand-over-hand stepping to move on single microtubules and between antiparallel microtubules, the authors need to revise this part of the Discussion and related Figure 7B-D to tone down the statement.

9) It is worth noting that the conclusions were drawn based on kinetic and structural studies using Dm-KLP61F (Figures 1 and 2) and other studies (TIRF microscopy and optical trapping) using human-Eg5 (Figures 3-6). I am unsure to what extent the authors can confidently claim the model in Figure 7 is a conserved one and strongly suggest the authors to use results from either Dm-KLP61F or human Eg5 for the main figures. Do the authors have results for Dm-KLP61F equivalent to those in Figures 3-6?

10) In terms of the organization of the manuscript, I would suggest the authors to present the figures in the following order: 1) Figures 3-5 (which show that the tail domain of kinesin-5s affects kinesin-5 velocity both on single MTs and within the overlap zone of two antiparallel microtubules, is required for efficient assembly of antiparallel microtubules, and is indispensable for kinesin-5 to produce strong forces within the overlap zone of two antiparallel microtubules), 2) Figures 1-2 (which are results showing the kinetic and structural bases of the observations in Figures 3-5), 3) Figure 6 (which shows that the tail domain plays important role in spindle localization of kinesin-5), and 4) the model Figure 7.

11) The authors state that "the tail domain of the *Xenopus* Eg5 was suggested to form a secondary [MT-binding] site during [MT-sliding] activity, yet the function of the kinesin-5 tail-MT interaction remains unclear." How does this extra MT-binding site fit into the model the authors propose in Figure 7?

---

## [Author Response]

Essential revisions:Reviewer 1:1) All figure captions, where applicable, need to state number of experimental repeats, type of statistics performed, how many molecules were included, units of average speed above histograms, axis units on inset graphs and brief experimental details. For example, Figure 1B-D, H, Figure 3F-G, Figure 4C-D, H, I, Figure 5D-F, Figure 6E.

These items have all been revised throughout the manuscript. These items are now included in the revised version in the figures or the tables and references were correctly added.

2) No model refinement parameters have been presented. Tables of refinement statistics is required for each model generated from the cryo-EM experiments.

We present Table 3 which includes model building and validation statistics in the revised manuscript.

3) FSC curve on Figure 2—figure supplement 1C(ii) shows a sharp drop-off (followed by a rise) at the low spatial frequency (~0.10 1/Å), usually indicative of an applied mask that is too tight. Can the authors try a looser mask?

We agree with the reviewer's point and thank them for this suggestion. The previous gold standard FSC calculation for final refinement was carried out using a home-built script. This has since been revised using the latest version of Frealign (CISTEM) which produced a revised FSC curve (shown in Figure 2—figure supplement 1) that doesn't show the sharp drop-off at low spatial frequency. The revised FSC curve now shows a normal drop off. We believe a final round of refinement with a mask that is less tight is not necessary. Masking the tail density does not improve the density suggesting that the variation in that region of the structure is quite high.

Reviewer 2:1) Exploring direct binding between the motor and tail domains:Motor-tail interaction is the key feature of this manuscript. The authors performed microtubule co-sedimentation assays as well as cryo-EM analysis to propose that the tail domain of KLP61F is not bound to the microtubule but is bound directly to the motor domain through an α-helix to regulate the catalytic cycle of the motor. However, biochemical assays did not show the direct binding between motors and tails without microtubules. Cryo-EM studies illustrate the additional density at the minus-end side of the motor, albeit it is much smaller than the expected size of the tail at ~8 Å resolution. Thus, it is not strong enough to conclude whether the additional density is actually the tail or not. The reviewers agree that a direct biochemical interaction between motor and tail should be tested without microtubules (e.g. by: GST-pull down assay, Gel-filtration (or SEC-MALS), Calorimetry, Biacore, etc).

To address the reviewers’ request and address this question directly, we generated a KLP61F tail with a C-terminal StrepII tag and utilized this construct to perform affinity co-purification of the KLP61F motor domain onto streptactin II resin, in three different nucleotide solution conditions. These studies are now included in Figure 1E. These studies reveal that the KLP61F tail domain recruited the motor domain in a stoichiometric amount to the streptactin resin (revealed by its elution with buffer with biotin) in the ADP and Nucleotide free states (Apyrase). The tail very poorly recruited the motor in the AMPPNP state. The ionic strength influences this interaction, which appears to become weaker at 75 mM KCl, compared to 25 mM KCl. Notably, size exclusion chromatography of the motor and tail shows they don't form a stable complex. This suggests that the interaction has an affinity in the micro-molar range.

2) Validating the motor-tail interface supposed by cryo-EM structure:Related to comment 1, a mutagenesis experiment will strengthen the structural model proposed. Please introduce a point mutation in α0-helix to investigate the contribution of this helix to the binding to the tail domain.

We thank the reviewer for this suggestion. However, given the very low resolution of the Cryo-EM map at tail region and motor-tail interface, It is very difficult identify the type of mutation in α0-helix that would disrupt the α-0 helix tail interface but won't affect the folding of α0-helix. The outcome of these mutation would not be easily validated if this region becomes unfolded or the interface of between the motor and tail becomes disrupted. We believe higher resolution structure for the motor tail interface is required before mutational analyses can be effectively used to test the biochemical activity of the tail-motor binding interface.

3) Checking the functional identity between KLP61F and Eg5:In the manuscript, the first half (biochemical and structural studies) was mostly done with Drosophila KLP61F, whereas the other half (biophysical studies) was done with human Eg5. The authors connect these stories together with the assumption that the role of the KLP61F and Eg5 are the same. Based on the sequence identity between KLP61F and Eg5, overall identity is 33%, and that of the motor domain is more than 50%, however, those of the tail and α 0 are around 20% , which is not high enough to easily conclude they play the same function. To overcome the gap, the authors need to experimentally prove functional identity between KLP61F and Eg5, including the motor-tail binding and the regulation of ATPase activity by tail under the same conditions (nucleotide analog, salt concentration, counter ion, etc.).

The kinesin-5 tail contains two regions of high conservation located at its N- and C-terminal sections. N-terminus of the tail (DM-KLP61F residues 930-970) is 80% conserved across mammals and *Drosophila*, while C-terminal section of the tail (Dm KLP61F residues 990-1020) is about 50% conserved. This suggests that about 50% or less of the tail is highly conserved, which is consistent with around 30% of the total mass of the 100 residues observed in cryo-EM. Our studies suggest that tail attributed density likely represents these conserved regions in a yet to be determined fold.

In response to the reviewer request of parallel biochemical analyses of Eg5 and KLP61F, we present ATPase and microtubule co-sedimentation studies with the Eg5 motor and tail domain constructs in Figure 1—figure supplement 1K-M of previous manuscript version. We have previously used ATP-γ-S which didn't mimic an effective ATP-like state in Eg5. In the revised version, we have now been expanded on these studies to address the above concerns of the reviewer. The co-sedimentation data in ATP-γ-S was removed and replaced with data collected in 2 mM AMPPNP.

As described above microtubule-stimulated ATP hydrolysis data show that the effect of the Eg5 tail construct is more moderate than KLP61F construct. We attribute this to the Eg5 tail construct solubility. Unlike the KLp61F tail construct, the isolated Eg5 tail construct suffers from soluble-aggregation issues, which likely influence the outcome of the motor + tail microtubule, stimulated ATPase assays. This is likely the reason for the more modest effect observed in these experiments. As we show below the Eg5 motor-tail fusion shows a more potent effect in decreasing the microtubule stimulated ATPase compared to the Eg5 motor alone likely due to the enhanced solubility. As we describe below, we studied this construct in two conditions to understand how the motor-tail interface affects the microtubule-activated ATP hydrolysis.

4) Decreasing the ATPase rate and the velocity with/without tail:The authors claim that the tail domain of kinesin-5 down-regulates the microtubule-activated ATPase activity of the motor, resulting in the slow motility of kinesin-5. In Table 1 and Figure 1, the ATPase rate of Eg5 was decreased to half by the addition of the tail. On the other hand, in Table 3 and Figure 3, the velocity of Eg5 was decreased to one-quarter or one-fifth by the addition of the tail. How can authors explain this discrepancy? Is there some additional reason of slow motility?

As we indicate above, the modest decrease in eg5-tail mediated microtubule-stimulated ATPase rate is likely due to the poor biochemical behavior of the Eg5 tail construct, which forms soluble aggregates over time. In the revised manuscript, we present ATP hydrolysis studies of the Eg5 motor-tail fusion compared to Eg5 motor at two ionic conditions 20 mM KCl and 50 mM K-acetate. We show that at 20 mM KCl conditions, the Eg5 motor tail fusion shows a two-fold decrease in Kcat ATP hydrolysis rate compared to the Eg5 motor, nearly identical to the effect observed with KLP61F. In contrast at 50 mM K-acetate condition we observe that the Eg5 motor ATPase decreases by 20% while its Km increased by seven-fold, while in contrast the Eg5 motor-tail fusion Kcat increases by two-fold compared to its kcat at 20 mM KCl. This suggests that the tail induced decrease in Eg5 motor ATPase is more sensitive to the 50 mM K Acetate condition. These data now provide a clearer view for how Eg5 tail ATPase regulation is altered and the role of the biochemical conditions and directly relates the ATPase rates to the motility studies described in Figure 3.

Reviewer 3:1) The authors need to significantly improve the clarity and accuracy of this manuscript in the revision, as the writing of the current version does not measure up to the quality of the work and the current manuscript contains surprisingly too many errors that have made the manuscript somewhat difficult to read. Some essential errors are listed below:1A) Panel E is missing in Figure 1; as a result, references to many panels of Figure 1 in the text are incorrect; and the text contains no mentioning of Figure 1J.1B) In the first paragraph of the subsection “The kinesin-5 tail domain downregulates MT activated ATP hydrolysis by binding the motor domain in the ADP or nucleotide-free states”, it should be "1-369" instead of "1-356", based on the labeling in Figure 1A.1C) "Figure 3A, lower panel" should be "Figure 3C".1D) “Figure 3—figure supplement 1C should be “Figure 3—figure supplement 1D-E”.1E) “Figure 3C” should be “Figure 3—figure supplement 1C”.

We are indebted to the reviewer for their thorough comments throughout our manuscript. The panels have now been corrected and the figure legends have been added.

2) Based on the GFP fluorescence intensity distribution of motile Eg5-Δtail-GFP and FL-Eg5-GFP, the authors state in the text (e.g. subsections “The tail domain down-regulates kinesin-5 motility velocity along single MTs”, third paragraph and “A modified model for kinesin-5 tail regulated hand-over-hand motility and its essential role in antiparallel MT sliding activity”, last paragraph) and in Figure 7A that in higher ionic strength buffers the tail of kinesin-5 may be involved in inter-molecular interaction to cluster multiple kinesin-5 homotetramers into higher-order oligomeric complexes. Given that in TIRF microscopy experiments, GFP fluorescence intensity of these motility Eg5 particles highly depends on experimental settings, e.g., laser intensity and location in the field of view, the authors need to use one other independent assay such as sucrose gradient centrifugation to verify whether FL-Eg5-GFP indeed exists mainly as high-order oligomeric complexes in high ionic strength buffers.

The imaging experiments of FL-Eg5-GFP and Eg5-tail-GFP at the 25 mM, 50 mM and 100 mM KCl condition were carried out side by side during same days while keeping control of the laser power and imaging conditions. Furthermore, data sets analyzed included imaging data merged from multiple days. Thus, potential differences in imaging conditions, indicated by the reviewer, do not explain the difference in oligomerization observed.

We attempted to address the reviewer's concern. We freshly purified FL-Eg5-GFP and Eg5-Δtail-GFP proteins at 400 mM KCl, then we incubated them at different ionic strengths in the absence of microtubules, and analyzed their oligomerization using gel filtration. However, we observed aggregation of both proteins at 25-100 mM KCl ionic strength conditions. In these experiments, protein concentrations are in the μM (μM) range, in contrast the TIRF experiments were carried out at 100-1000-Fold lower concentration. We believe that the microtubule lattice plays an important role in the clustering process in the nanomolar (nM) range, as FL-Eg5-GFP molecules assemble into these oligomers on the microtubule lattice.

*3) While the authors have demonstrated that the tail domain slows kinesin-5 velocity both along single MTs and within the overlap zone formed between two antiparallel MTs, it is unclear whether this tail-to-motor interaction indeed forms part of an active regulatory pathway* in vivo*. As such, in the Results, the authors need change "downregulate" and "down-regulate" (or their likes) to reduce/decrease to focus on describing rather than interpreting their experimental observations.*

This has been corrected.

4) In the second paragraph of the Introduction, the authors state that yeast kinesin-5s reverse direction toward the plus-ends upon clustering into multi-motor complexes along single MTs. Which study (or studies) are the authors referring to?

The work by Saphira et al., 2017, describes the studies on yeast Cin8 direction change along microtubules. The reference has been added in the revised manuscript.

5) It is inaccurate to state that in motility buffers contains 25-100mM KCl, "[t]his processive motility is highly homogeneous", given that the velocity histograms for FL-Eg5-GFP and Eg5-∆tail-GFP at 100 mM KCl both contain two different modes.6) In the subsection “The tail domain is critical for slowing kinesin-5 motility within MT sliding zones”, the description of FL-Eg5-GFP motility is inaccurate, and it is not obvious from Figure 4F that FL-Eg5-GFP exhibits slower plus-end directed motility within the MT overlap zone than on the surface-anchored MT outside the overlap zone; judging by the inset velocity histograms in Figure 4H, it is unclear whether the difference between the two velocities is statistically significant.

Statistical significance for the data is now presented in Figure 4—figure supplement 1F-G as a pairwise student's t-tests

7) In the subsection “The tail domain is critical for slowing kinesin-5 motility within MT sliding zones”, it is not clear to me how these data would necessarily suggest the kinesin-5 tail regulates MT-sliding rate by modulating a unique motor association with MTs and coupling between two MT-bound ends of kinesin-5. How about the presence of the tail domain prolonging the retention of kinesin-5 within the MT-binding overlap zone by being an extra MT-binding site?

The spiking experiments clearly indicate that the velocity of the FL-Eg5 motor within the overlap zones is two-three folds slower than the Eg5-Δtail-GFP. The Fl-Eg5-GFP motors are not static but rather moving slowly, in contrast to the Eg5-Δtail-GFP move rapidly with very frequent direction switching within the overlap zone. Furthermore, kinetic and structure data in Figures 1 and 2 show that the affinity of the tail to the motor domain is highest in the ADP and nucleotide free states. The decrease in microtubule stimulated ATP hydrolysis in the presence tail and the location of the tail binding site on the motor domain both suggest that the tail interface stabilizes the nucleotide free or ADP states of the motor. The data suggest that the explanation for the very slow motility velocity of FL-Eg5-GFP motor domain in the overlap zone is likely due to its slow transition due to tail inducing a decreased affinity for ATP, the transition from ADP state to ATP state by stabilizing the nucleotide free or ADP states.

8) In the Discussion, the authors state in multiple places that the tail domain regulates processive hand-over-hand stepping during kinesin-5 motility. Given that it remains to be determined whether homotetrameric kinesin-5s use hand-over-hand stepping to move on single microtubules and between antiparallel microtubules, the authors need to revise this part of the Discussion and related Figure 7B-D to tone down the statement.

In the data presented in Figure 3, we show that Eg5 tetramers undergo processive motility along microtubules with run lengths that are between 8 and 13 micro-meters (μm). We also would like to point the reviewer to the work of Valentine et al., 2005, on native dimeric Eg5 constructs which include the N-terminal neck, which unambiguously show these molecules take steps similar to kinesin-1 using optical trapping. The conclusion that each dimeric end of Eg5 takes processive steps is fairly reasonable based on our data and that of the Valentine et al. work.

9) It is worth noting that the conclusions were drawn based on kinetic and structural studies using Dm-KLP61F (Figures 1 and 2) and other studies (TIRF microscopy and optical trapping) using human-Eg5 (Figures 3-6). I am unsure to what extent the authors can confidently claim the model in Figure 7 is a conserved one and strongly suggest the authors to use results from either Dm-KLP61F or human Eg5 for the main figures. Do the authors have results for Dm-KLP61F equivalent to those in Figures 3-6?

We have revised the Discussion in the manuscript. We include additional data addressing Eg5 tail capacity to regulate the motor and added data with an Eg5 motor-tail fusion to address how the tail regulates ATPase at low and higher ionic strength conditions. We believe the sequence conservation in the motor and tail domains (see above), reasonable match in the biochemical microtubule binding and microtubule-stimulated ATPase assays for the Eg5 and KLP61F motor and tail domains, cryo-EM KLP61F structures revealing structural basis for the tail effect on the motor domain, and the reconstitutions of the tail effect in the FL-Eg5 and Eg5-Δtail microtubule motility and sliding studies all support that these features are likely conserved in the Eg5 and KLP61F motor and tail domains.

10) In terms of the organization of the manuscript, I would suggest the authors to present the figures in the following order: 1) Figures 3-5 (which show that the tail domain of kinesin-5s affects kinesin-5 velocity both on single MTs and within the overlap zone of two antiparallel microtubules, is required for efficient assembly of antiparallel microtubules, and is indispensable for kinesin-5 to produce strong forces within the overlap zone of two antiparallel microtubules), 2) Figures 1-2 (which are results showing the kinetic and structural bases of the observations in Figures 3-5), 3) Figure 6 (which shows that the tail domain plays important role in spindle localization of kinesin-5), and 4) the model Figure 7.

We deeply appreciate the reviewers suggested reorganization scheme. However, we have previously attempted to organize a previous draft of the manuscript in this exact manner and the results were more difficult to interpret for an audience of readers. In the current format, the biochemical and structural data allow a straightforward interpretation of the functional data presented after the biochemical and structural studies leading to a much clearer manuscript.

11) The authors state that "the tail domain of the Xenopus Eg5 was suggested to form a secondary [MT-binding] site during [MT-sliding] activity, yet the function of the kinesin-5 tail-MT interaction remains unclear." How does this extra MT-binding site fit into the model the authors propose in Figure 7?

Our study suggests the previous work by Weinger et al., 2009, on similar versions of *Xenopus* Eg5 constructs should be re-interpreted in light of our data. Our studies show that the kinesin-5 tail primary binding site is located on the kinesin-5 motor domain and not on the microtubule lattice. However, we cannot rule out a secondary binding site of the tail domain on the microtubule lattice but believe the main effect comes from its motor bound form. Our data show that the tail domain binding on the motor likely regulates the ATP hydrolysis by stabilizing the nucleotide free or ADP-states, leading the motor to exhibit slow motility along microtubules particularly within overlap zones, as shown in Figure 4.